# Energetic and durable all-polymer aqueous battery for sustainable, flexible power

Yang Hong[1,2,10], Kangkang Jia[1,10], Yueyu Zhang [ID]3, Ziyuan Li[4], Junlin Jia[5], Jing Chen[6], Qimin Liang[1], Huarui Sun [ID]1, Qiang Gao[6], Dong Zhou [ID]7, Ruhong Li [ID]8, Xiaoli Dong [ID]9, Xiulin Fan [ID]8 ✉ & Sisi He [ID]1 ✉

All-polymer aqueous batteries, featuring electrodes and electrolytes made entirely from polymers, advance wearable electronics through their processing ease, inherent safety, and sustainability. Challenges persist with the instability of polymer electrode redox products in aqueous environments, which fail to achieve high performance in all-polymer aqueous batteries. Here, we report a polymer-aqueous electrolyte designed to stabilize polymer electrode redox products by modulating the solvation layers and forming a solid-electrolyte interphase. Polyaniline is selected as an example for its dual functionality as a cathode or anode working by p/n doping mechanisms. This approach pioneers the application of polyaniline as an anode and enhances the high-voltage stability of polyaniline cathode in an aqueous electrolyte. The resulting all-polymer aqueous sodium-ion battery with polyaniline as symmetric electrodes exhibits a high capacity of 139 mAh/g, energy density of 153 Wh/kg, and a retention of over 92% after 4800 cycles. Spectroscopic characterizations have elucidated the hydration structure, solid-electrolyte interphase, and dual-ion doping mechanism. Large-scale all-polymer flexible batteries are fabricated with excellent flexibility and recyclability, heralding a paradigmatic approach to sustainable, wearable energy storage.

Flexible and safe batteries have recently gained escalating attention with the rapidly growing demands of wearable technologies[1–3]. Although lithium-ion batteries have dominated portable electronics for decades, their reliance on mineral resources and flammable/toxic organic electrolytes highlights safety and sustainability challenges[4–8]. Aqueous sodium-ion (Na-ion) batteries (ASIBs) emerge as a more sustainable alternative to lithium-ion batteries due to their abundant sources and intrinsic safety[9–11]. However, the lack of suitable electrode

materials for ASIBs still restricts their practical applications with limited energy density and poor cycling stability[12].

Organic electrode materials are expected to be promising substitutes for inorganic electrode materials in ASIBs due to their rich structural designability, high capacity and sustainability[13–15]. Unfortunately, restricted by the low redox potential of organic electrode materials, only a few carbonyl derivatives successfully serve as anodes, and the cathode materials are still almost confined within activated

[1]School of Science, Harbin Institute of Technology (Shenzhen), Shenzhen, China. [2]Department of Chemistry and Biotechnology, School of Engineering, The University of Tokyo, Tokyo, Japan. [3]Wenzhou Institute University of Chinese Academy of Sciences, Wenzhou, China. [4]School of Physical Science and Technology, Ningbo University, Ningbo, Zhejiang, China. [5]School of Physics, East China University of Science and Technology, Shanghai, China. [6]School of Chemistry and Chemical Engineering, Yangzhou University, Yangzhou, China. [7]Tsinghua Shenzhen International Graduate School, Tsinghua University, Shenzhen, China. [8]State Key Laboratory of Silicon and Advanced Semiconductor Materials, School of Materials Science and Engineering, Zhejiang University, Hangzhou, China. [9]Department of Chemistry and Shanghai Key Laboratory of Molecular Catalysis and Innovative Materials, Institute of New Energy, iChEM (Collaborative Innovation Center of Chemistry for Energy Materials), Fudan University, Shanghai, China. [10]These authors contributed equally: Yang Hong, Kangkang Jia. ✉e-mail: xlfan@zju.edu.cn; hesisi@hit.edu.cn

carbon and other inorganic materials, whose capacity and cycling stability are still inadequate[16–20]. On the other hand, current electrodes in flexible aqueous alkali-metal-ion batteries are constrained to a few inorganic materials, and most of those batteries are Li-ion batteries, which exacerbates the sustainability concerns[21–27]. Moreover, the development of flexible electronics is based on the rise of polymer science[28]. Hence, considering the requirements of flexibility and processability for the application in flexible batteries, all-polymer ASIBs based on polymer electrode materials are urgently awaited[29].

Herein, we report an energetic and flexible all-polymer ASIB using polyaniline (PANI) as symmetric electrodes with a polymer-aqueous electrolyte (PAE). PANI is one of the most well-studied polymer electrode materials, characterised by its low cost, easy synthesis, and multiple redox states[30–33]. Generally, pristine PANI is half-oxidised and can be oxidised/reduced via anion/cation doping[34,35]. Thus, PANI shows the potential as symmetric electrodes for assembling all-polymer flexible batteries (Fig. 1a). Besides, the simple chemical structure of PANI also provides us with the most typical example for understanding the working mechanism of all-polymer aqueous batteries. However, due to the preconception that the negatively charged nitrogen atoms (nitrenes) of the metal pernigranilate (MPN) are unstable under protic conditions, there is no reported cation-dopped PANI anode in an aqueous electrolyte[36]. Hence, almost all aqueous devices based on symmetric PANI electrodes are supercapacitors without evident redox peaks and efficient doping. To break this longstanding preconception and attain ASIBs with PANI as symmetric electrodes, success hinges on electrolyte design. Specifically, polyethylene glycol dimethyl ether's (PEGDME) ability to modulate hydration layers and form solid-electrolyte interphase are paramount.

PANI and carbonyl derivatives are the most well-studied organic cathode and anode materials in aqueous batteries. Initially, we assembled a series of ASIBs based on PANI and carbonyl derivatives as a pre-experiment for developing all-polymer ASIBs (Supplementary Table 1). Synthesis and characterisation of PANI and poly-(naphthalene four formyl ethylenediamine) are described in the supplementary information (Supplementary Figs. 1, 2). Although PANI is widely used as a cathode material, the oxidised product (pernigraniline salt, PNS) offers strong electrophiles to water[37,38]. It is easily deprotonated, leading to poor cycling stability in neutral aqueous electrolytes (Fig. 1b). Inspired by the potential to extend the electrochemical stability window of advanced aqueous electrolytes through modulation of cation hydration structures and the formation of stable electrolyte-electrode interphases, we proposed to reduce the activity of $H_2O$ for stabilizing highly oxidised products of PANI electrodes. The high salt concentration is an efficient method for improving the stability of aqueous electrolytes, but it may bring concerns about high weight and potential corrosivity when applied in wearable electronics. To satisfy the requirements of flexibility, portability, and sustainability of flexible batteries, we chose PEGDME as the modulator for controlling the activities and dynamics of $H_2O$ molecules due to its advantages of low cost, low density, low volatility, and high biocompatibility. However, although the electrochemical stability window (ESW) could be widened to 3.2 V in this PAE, the ASIBs based on carbonyl derivates and conducting polymers didn't deliver the expected performances (Supplementary Figs. 3–5). Fortunately, we found that the half-cell assembled with PANI and active carbon exhibits an energy density of 142 mAh/g (Supplementary Fig. 6). Thus, we confirm that the deprotonation of high oxidised PANI cathode can be suppressed due to the decreased $H_2O$ activities in the PAE. Then, considering the dual-mode (anion/cation) doping potential of PANI, we attempted to create ASIBs with PANI as symmetric electrodes based on this PAE. Unexpectedly, n-doping of PANI was achieved in this PAE (Fig. 1c). More significantly, the assembled symmetric PANI ASIBs exhibit a high capacity of 139 mAh/g and an energy density of 153 Wh/kg, with

a high retention of 92.0% after more than 4800 cycles and an average coulombic efficiency of 99.5%.

This all-polymer ASIB shows high energy density and cycling stability far surpassing that of those ASIBs assembled with PANI and other commonly used organic anode materials (Fig. 1d) and is even good among the most advanced ASIBs (Fig. 1e). Moreover, this all-polymer ASIBs possesses sustainability because of the use of totally metal-free active materials and mild-concentration (2 m) electrolyte compared with other high-performance ASIBs[20,39–42]. The all-polymer ASIB also demonstrates extraordinary processability, flexibility, and recyclability. This opens an avenue for developing high-performance all-polymer aqueous batteries for sustainable and flexible power technologies. More importantly, the rich molecular engineering methods of polymer materials will boost the development of multi-functional flexible batteries, such as stretchable, self-healable and shape-memory devices.

## Results

### Electrolyte design and solvation structure

Compared with conventional aqueous electrolytes, the ESW of PAE can be expanded to 3.2 V without using high-concentration sodium salt (Fig. 2a). To investigate the effect of PEGDME on the electrochemical properties of PAE, a series of 2 m $NaTFSI$-PEGDME-$H_2O$ solutions were studied here with a comparison with 2 m $NaTFSI$-$H_2O$ solution. First, we investigated the ESW and ionic conductivity of different PAEs by adjusting the weight-average molecular weights of PEGDME from 250 to 500 g/mol. As the molecular weight increased, the ESW widened, but ionic conductivity decreased (Supplementary Figs. 7, 8). Besides, the salt concentration dependencies of the ionic conductivities and the ESW of PAE were also studied (Supplementary Fig. 9). With the salt concentration increasing from 2 m to 5 m, the ionic conductivities decreased due to the sharply increasing viscosity. The ESW of 4 m $NaTFSI$-PAE also didn't show an obvious increment compared to 2 m $NaTFSI$-PAE. Apart from PEGDME, we found other ethers-containing polymers like polypropylene glycol (PPG) also can enlarge the electrochemical stability of aqueous electrolytes but showing lower ionic conductivities (Supplementary Fig. 10). Therefore, we chose 2 m $NaTFSI$-PEGDME(450)-$H_2O$ solution as the PAE in our studies. Proton nuclear magnetic resonance spectroscopy ($^{1}H$-NMR), Fourier-transform infrared spectroscopy (FTIR), and Raman spectroscopy were applied to study the effect of PEGDME on the $H_2O$ molecules and solvation layers structures in the electrolyte. $^{1}H$-NMR spectroscopies of 2 m $NaTFSI$-PAE, 2 m $NaTFSI$-$H_2O$ electrolytes, and $H_2O$ were performed in a deuterated DMSO solution (Fig. 2b). With the increasing amount of PEGDME, the $^{1}H$ chemical shift of $H_2O$ moved upfield, indicating a higher electron density around the H atom of $H_2O$. It is usually due to a shorter H-bond between donor and acceptor inducing more shielding of H, and such changes demonstrate the breaking of the H-bond network among $H_2O$ molecules and the formation of a stronger H-bond between $H_2O$ and ether chain. FTIR spectroscopy was further utilized to characterise H−O stretching and bending of $H_2O$ in different electrolytes (Fig. 2c). The typical H−O stretching and bending modes of $H_2O$ are 3300−3500 $cm^{-1}$ and 1638 $cm^{-1}$, respectively, which shows less change in 2 m $NaTFSI$ but an apparent blue shift in 2 m $NaTFSI$-PAE (to 3400−3600 $cm^{-1}$ and 1652 $cm^{-1}$ respectively). Such changes in H−O stretching and bending further prove that the hydrogen-bond (H-bond) network in bulk $H_2O$ was broken by adding PEGDME[43,44].

Compared with $H_2O$ and 2 m $NaTFSI$-$H_2O$, the Raman spectrum of 2 m $NaTFSI$-PAE shows that the symmetric stretching mode (3249 $cm^{-1}$) of strong tetrahedral H-bonded (DDAA, D refers to H-bond donor, A refers to H-bond acceptor) bulk $H_2O$ disappeared, only the asymmetric stretching modes (3420 $cm^{-1}$, 3572 $cm^{-1}$) of asymmetric H-bonded (DDA and DA) $H_2O$ molecules, and symmetric stretching mode (3636 $cm^{-1}$) of free $H_2O$ can be observed (Fig. 2d)[45,46]. To clarify

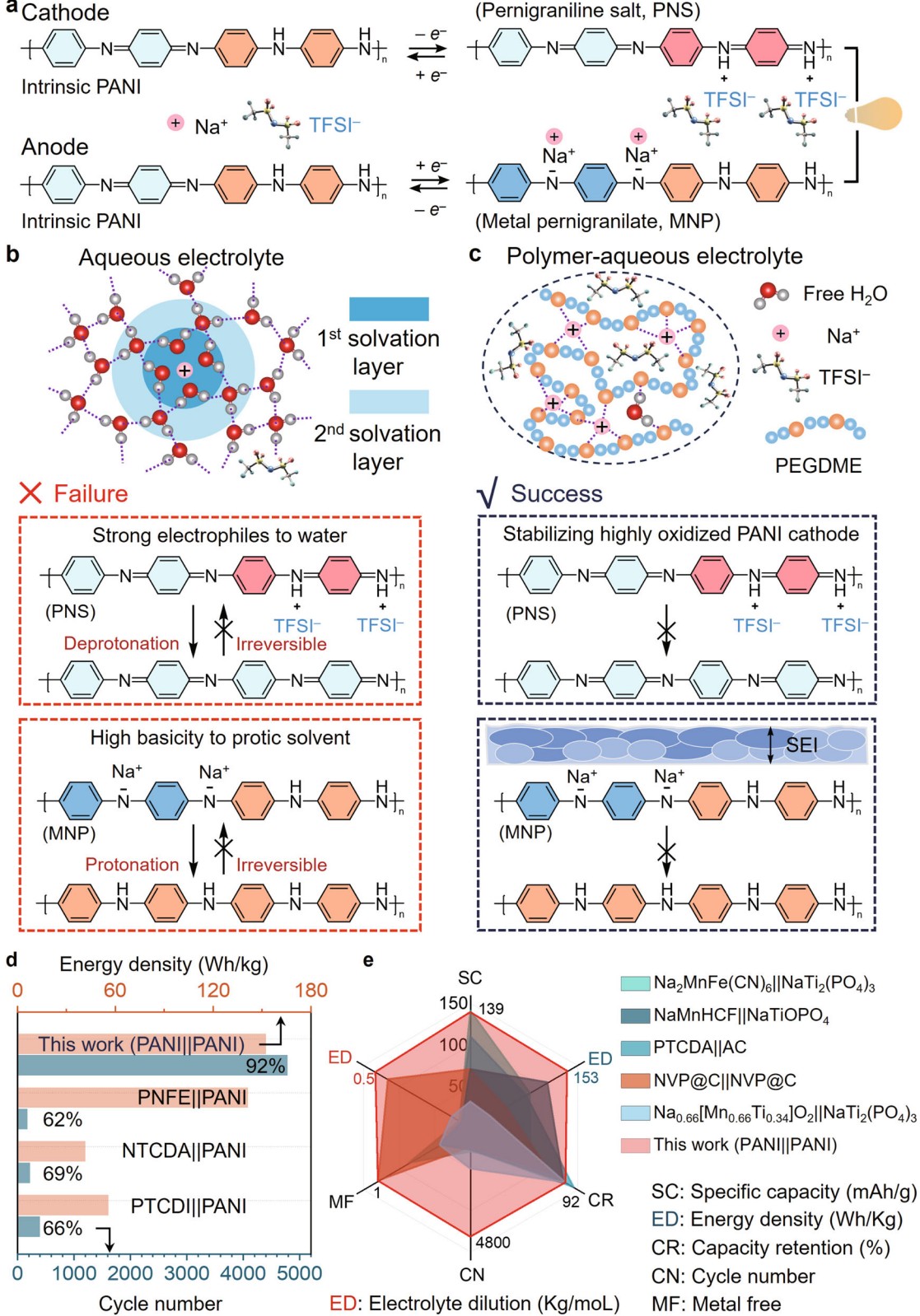

**Fig. 1 | Schematic figure of all-polymer aqueous sodium-ion batteries enabled by polymer-aqueous electrolyte. a** Schematic figure of the dual-ion doping mechanism of symmetric PANI electrodes during charge. **b** Failure mechanism of symmetric PANI electrodes in 2 m NaTFSI aqueous electrolyte. **c** Stable all-polymer ASIBs enabled by 2 m NaTFSI polymer-aqueous electrolyte. **d** Comparison with other all-organic ASIBs assembled with carbonyl derivatives anodes and PANI cathodes. **e** Specific capacity (SC), energy density (ED in blue), capacity retention (CR), cycle number (CN), metal-free (MF), and electrolyte dilution (ED in red) of this work in comparison with advanced ASIBs. Both specific capacity and energy density are calculated based on the mass of active materials in cathodes. MF, all organic electrodes: 1; single organic electrode: 0.5; all inorganic electrodes: 0.

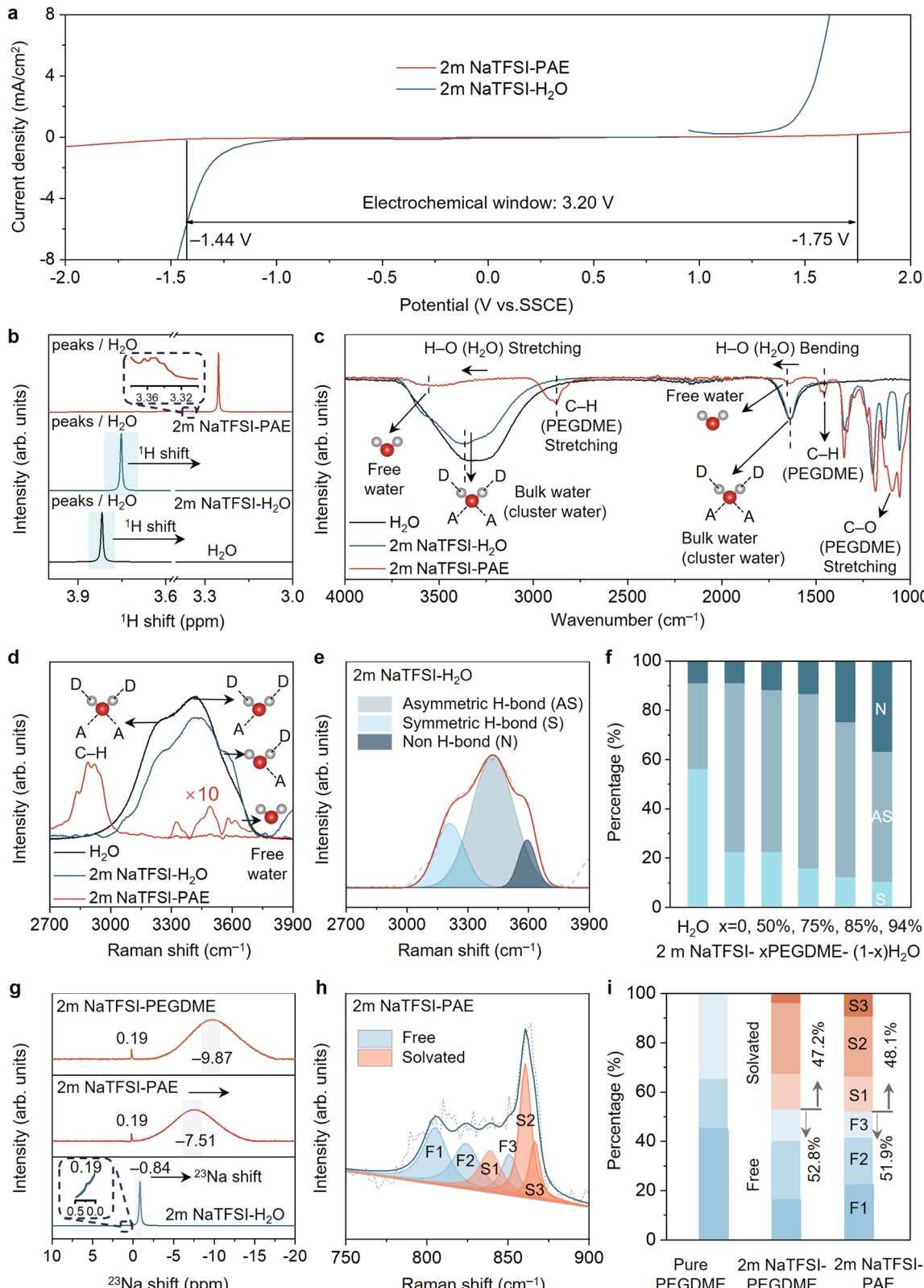

**Fig. 2 | Electrolyte design and hydration structure. a** Electrochemical stability window of 2 m NaTFSI-$H_2O$ and 2 m NaTFSI-PAE. **b** Proton NMR spectra of $H_2O$, NaTFSI-$H_2O$ and 2 m NaTFSI-PAE. **c** FTIR spectra of $H_2O$, 2 m NaTFSI-$H_2O$ and 2 m NaTFSI-PAE. **d** Raman spectra of $H_2O$, 2 m NaTFSI-$H_2O$ and 2 m NaTFSI-PAE. **e** Raman peak fittings of the O–H stretching vibrations with different hydrogen-bond environments, including strong, weak, and non-hydrogen bonds.

**f** Percentage of fitted area for $H_2O$ with the symmetric H-bonded (S) asymmetric H-bonded (AS), and non-H-bonded (N) $H_2O$ with increasing PEGDME concentration in 2 m NaTFSI-PAE. **g** Sodium NMR spectra of 2 m NaTFSI-PEGDME, 2 m NaTFSI-PAE, and 2 m NaTFSI-$H_2O$. **h** Raman peak fittings of the C–O–C stretching vibrations within/without solvation layers. **i** Percentage of fitted area for ether groups within/without solvation layers.

the effect of PEGDME on the H-bond network of $H_2O$, Raman spectra with different PEGDME concentrations (50, 75, 85, 94% in weight) were further conducted (Supplementary Fig. 11), which show an apparent blue shift of symmetric stretching mode (3315 $cm^{-1}$ to 3340 $cm^{-1}$) with increasing PEGDME concentrations. Then, we evaluated the $H_2O$ structure in different H-bond states by calculating the percentage of each component based on the area of fitted peaks (Fig. 2e, and Supplementary Fig. 12). In Fig. 2f, the percentages of the asymmetric H-bonded and non-H-bonded $H_2O$ molecules increased with increasing PEGDME concentrations, indicating the breaking of H-bond network of bulk $H_2O$ and the generation of free $H_2O$[47,48]. Compared with H-bonding bulk $H_2O$, the non H-bonding free $H_2O$ shows lower dielectric constants and polarities, which is more difficult to be electrolysed[49]. Besides, it is apparent that in PAE electrolyte, the asymmetric H-bonded $H_2O$ molecules become the dominant component (53%–69%) compared with 2 m NaTFSI, which further proves that most $H_2O$ molecules are forming asymmetric H-bonded with PEGDME. All the above characterisation results demonstrate the strength enhancement of H−O bonds and the reduced activity of $H_2O$ in PAE electrolyte[37]. The solvation layer structures of $Na^+$ were further studied by sodium nuclear magnetic resonance spectroscopy ($^{23}Na$ NMR), and Raman spectra. The $^{23}Na$ NMR spectra show that the chemical shifts of $^{23}Na$ in 2 m NaTFSI-$H_2O$, and 2 m NaTFSI-PEGDME are around − 0.84 ppm and − 9.87 ppm, respectively. The $^{23}Na$ shift of 2 m NaTFSI-PAE is − 7.51 ppm, which is quite near 2 m NaTFSI-PEGDME but far away from 2 m NaTFSI-$H_2O$, indicating the main component of $Na^+$ solvation layer is PEGDME. However, the slight movement of $^{23}Na$ in 2 m NaTFSI-PAE to the downfield compared with 2 m NaTFSI-PEGDME, such deshielding effect illustrates that $H_2O$ is also involved in the solvation of $Na^+$ in 2 m NaTFSI-PAE (Fig. 2g). The solvation of $Na^+$ with the ether group also can be proved by the redshift of C-O-C stretching in 2 m NaTFSI-PAE when compared with PEGDME. Then, we evaluated the ratio of ether groups in $Na^+$ solvation layers in 2 m NaTFSI-PAE by calculating the percentage of ether groups in different chemical environments, based on the area of fitted peaks of Raman spectra (Fig. 2h and Supplementary Fig. 13). Taking the solvated ether ratio in 2 m NaTFSI-PEGDME as the reference, the fitting results also clearly prove that PEGDME is the main component of the $Na^+$ solvation layer in 2 m NaTFSI-PAE (Fig. 2i). The slightly decreased ratio of solvated ether also proves that there are small amounts of $H_2O$ in the $Na^+$ solvation layers, which can efficiently improve the ionic conductivity of 2 m NaTFSI-PAE compared with 2 m NaTFSI-PEGDME (Supplementary Fig. 14). Based on this electrochemically stable and high conductive PAE electrolyte, further studies on the electrochemical behaviour of PANI can be conducted on a broader voltage range.

## Symmetric PANI battery with dual-ion doping mechanism

Although intrinsic PANI shows a half-oxidised state and the potential of serving as a cathode or anode through reversible anion or cation doping, there is no report that PANI works as an anode in aqueous electrolyte due to the preconception that n-doping of PANI cannot be achieved in the protic solvent[36]. To prove the possibility of constructing all-polymer ASIB with PANI as symmetric electrodes, first-principles density functional theory (DFT) simulations, cyclic voltammetry measurements and spectral characterisation were utilized. The DFT simulation shows that intrinsic PANI could gain two electrons via $Na^+$ doping or lose two electrons with TFSI$^-$ doping (as shown in the scheme in Fig. 3a). The thermal Gibbs free energy changes of the PANI cathode and anode throughout the entire charging process are shown in Fig. 3b. The secondary amine of intrinsic PANI can be easily doped with TFSI$^-$ through a two-step spontaneous reaction process with a Gibbs free energy decrease of − 5.88 eV and the loss of two electrons. As for the anode, the imine of intrinsic PANI consumes 7.48 eV to gain two electrons via $Na^+$ doping. The symmetric PANI battery system consumes 1.57 eV for a two-

electron-transfer electrochemical process throughout the dual-ion doping process. The calculated theoretical specific capacity based on the two-electron transfer of symmetric PANI batteries is about 147 mAh/g. Then, three-electrodes measurements of the cathode and anode were conducted with PANI as the working electrode between voltage windows of 0–1.0 V and − 1.0-0 V, respectively (Supplementary Fig. 15). The practical capacities of PANI cathode and anode are 135 mAh/g and 130 mAh/g in the three-electrode system, respectively. All-polymer aqueous batteries were assembled with symmetric PANI electrodes. Since symmetric conductive polymer electrodes typically work as a supercapacitor in aqueous electrolytes, cyclic voltammetry measurements were utilized to investigate the kinetics of electrode reaction (Supplementary Fig. 16). Generally, the relationship between response current and sweep rates follows the empirical Eq. (1)[50]:

$$i = av^b \tag{1}$$

where ($i$) is peak current, ($v$) is potential sweep rate, and ($a$ and $b$) are fitting parameters. When the $b$ value is around 0.5, the charge storage process is controlled by ion diffusion, indicating the device is an electrochemical battery; when the $b$ value is around 1, the charge storage process relies on surface absorption, indicating the device is a capacitor. The $b$ value of this all-polymer aqueous device is ∼ 0.75, proving that our device includes both diffusion and absorption working mechanisms. For further determining the working mechanisms, diffusion/capacitance contributions at different scan rates are calculated with another empirical equation[51]:

$$I(v) = k_1v + k_2v^{0.5} \tag{2}$$

where $I(v)$ is peak current, $v$ is potential sweep rate, $k_1$ and $k_2$ are fitting parameters. The $k_1v$ part relies on surface absorption on the electrodes, indicating the capacitance contribution *and the $k_2v^{0.5}$* part is controlled by ion diffusion inside electrodes, indicating the battery storage behaviour. The ratio of diffusion and capacitance contributions in this all-polymer aqueous device under different scan rates was calculated (Supplementary Fig. 16c). The results show that when the scan rate is below 1 mV/s, diffusion behaviour is always dominant. Hence, we can achieve all-polymer batteries instead of general capacitors, boosting the research interest in polymer-based high-energy aqueous devices.

To elucidate the actual dual-ion doping process of each electrode in a symmetric PANI battery, FTIR, Raman and X-ray photoelectron spectroscopy (XPS) were employed to track the redox state of both PANI electrodes under different charge-discharge potentials in the initial two cycles. From the FTIR spectra of the cathode, it is evident that typical stretching modes of S–N–S (879 $cm^{-1}$), O = S = O (1051 $cm^{-1}$), and $CF_3$ (1211 $cm^{-1}$) periodically occur during charging and disappear during discharging (Fig. 3c and Supplementary Fig. 17). Raman spectra of PANI cathode also shows the intensity of C–N$^+$ stretching (1335 $cm^{-1}$) enhanced with charging and weaken with discharging (Supplementary Fig. 18). These results confirm the oxidation of PANI when TFSI$^-$ doping happening[52]. The reversible doping of TFSI$^-$ on the PANI cathode is also observed in XPS F$1s$ spectra (Fig. 3d). The initial peak of F$1s$ of pristine cathode originates from the polytetrafluoroethylene (PTFE) binder in the cathode. With the potential charging to 2.2 V, the intensity of F$1s$ increases significantly due to the doping of TFSI$^-$ onto the PANI cathode, which returns near the initial value after the potential discharging to 0.1 V due to the dedoping of TFSI$^-$. Such doping and dedoping of anion repeat during the second cycle, proving the reversibility of TFSI$^-$ doping onto the PANI cathode between 0.1–2.2 V. For the anode, FTIR spectra show the reversible shift of C = N (1290 $cm^{-1}$) in the pristine PANI to the C = N$^-$ (1305 $cm^{-1}$) in the reduced PANI during the charge-discharge cycles (Fig. 3e and

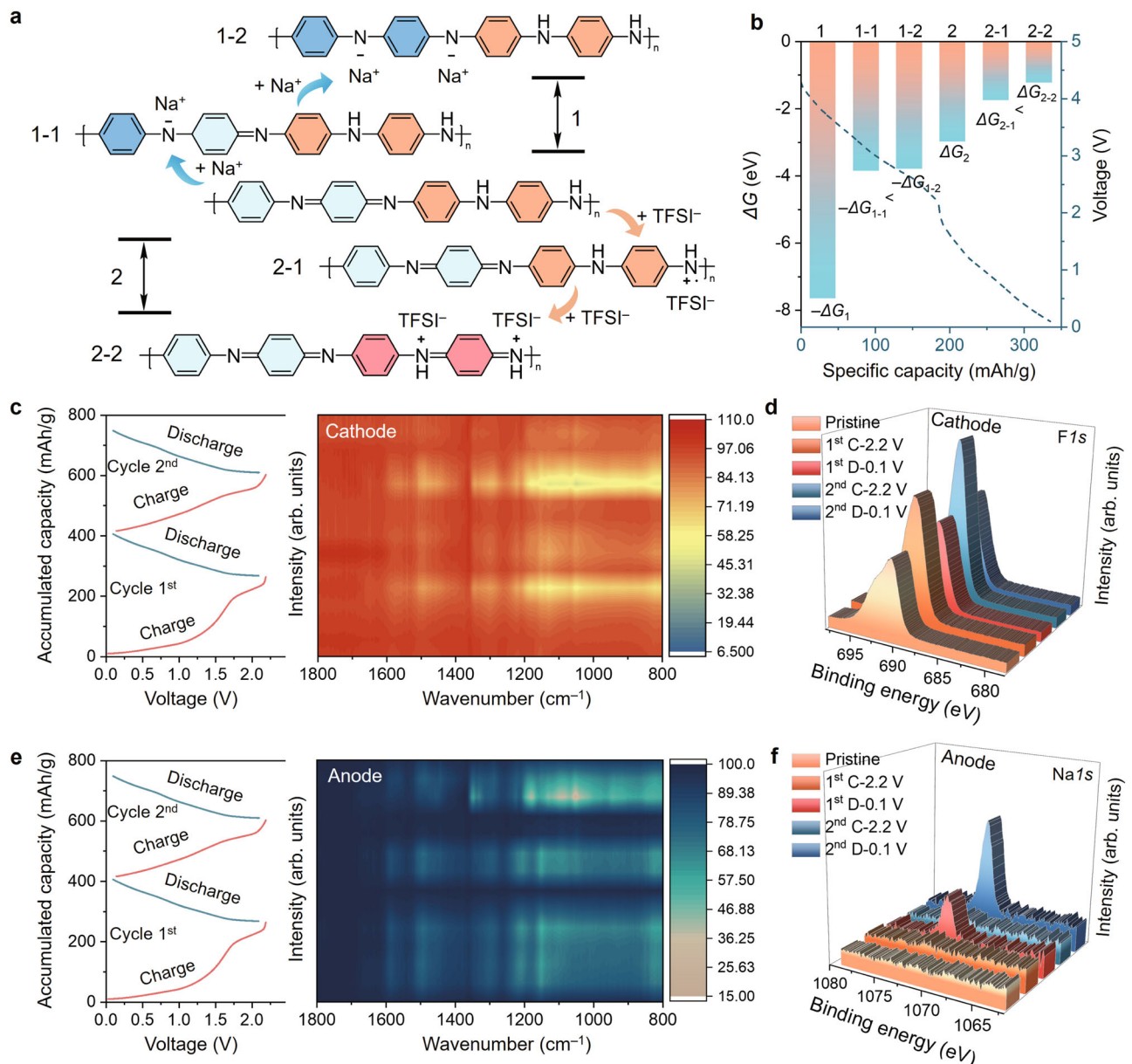

**Fig. 3 | Dual-ion doping mechanism of all-polymer aqueous sodium batteries with symmetric PANI electrodes. a** Schematic figure of the dual-ion doping mechanism of symmetric PANI electrodes during charge. **b** Charge potential in the experiment and Gibbs free energy changes in the stimulation of dual-ion doping in all-polymer batteries with PANI as symmetric electrodes. **c** FTIR spectra of PANI cathode at different potentials in two charge-discharge cycles. **d** XPS spectra of PANI cathode at different potentials in two charge-discharge cycles. **e** FTIR spectra of PANI anode at different potentials in two charge-discharge cycles. **f** XPS spectra of PANI anode at different potentials in two charge-discharge cycles.

Supplementary Fig. 19). XPS Na$1s$ spectra were further applied to characterise the contribution of Na$^+$ in the anode reaction, Fig. 3f also shows a reversible change of Na$1s$ intensity during two charge-discharge cycles. The spectroscopy under different potentials proves that PANI works as symmetric electrodes in aqueous batteries based on the reversible TFSI$^-$ and Na$^+$ doping on the PANI cathode and anode, respectively.

## Stable all-polymer battery with interface chemistry

Encouraged by the reversible charge-discharge behavior of PANI in PAE, we fabricated the all-polymer battery using PANI as the symmetric electrodes with 2 m NaTFSI-PAE. Electrochemical performances of all-polymer ASIBs were further studied using galvanostatic charge-discharge (GCD) and cyclic voltammetry (CV). The all-polymer battery can deliver a high specific capacity of 139 mAh/g and an energy density of 153 Wh/kg at a 1 C rate (Fig. 4a). More remarkably, it can maintain 92.0% capacity after 4800 cycles (381 days), with a high average coulombic efficiency (CE) of 99.5% (Fig. 4b), which is much more stable than most ASIBs and aqueous Li-ion batteries (Supplementary Table 4). It proved that our strategy to stabilize highly oxidised/reduced PANI with PAE works well. CV measurements further revealed the difference in the redox process of the symmetric PANI electrodes in the 2 m NaTFSI-PAE and 2 m NaTFSI. The anodic peak at ~1.8 V was observed in the first cycles in 2 m NaTFSI-PAE, which could be attributed to forming a solid electrolyte interphase. CV measurements of other all-organic batteries in 2 m NaTFSI-PAE confirm that the anodic peak does not depend on the electrode species (Supplementary Figs. 20, 21). After the first cycle, the symmetric PANI batteries exhibited a reversible redox reaction of PANI around 1.2 V without any other side reactions between 0.1–2.2 V (Supplementary

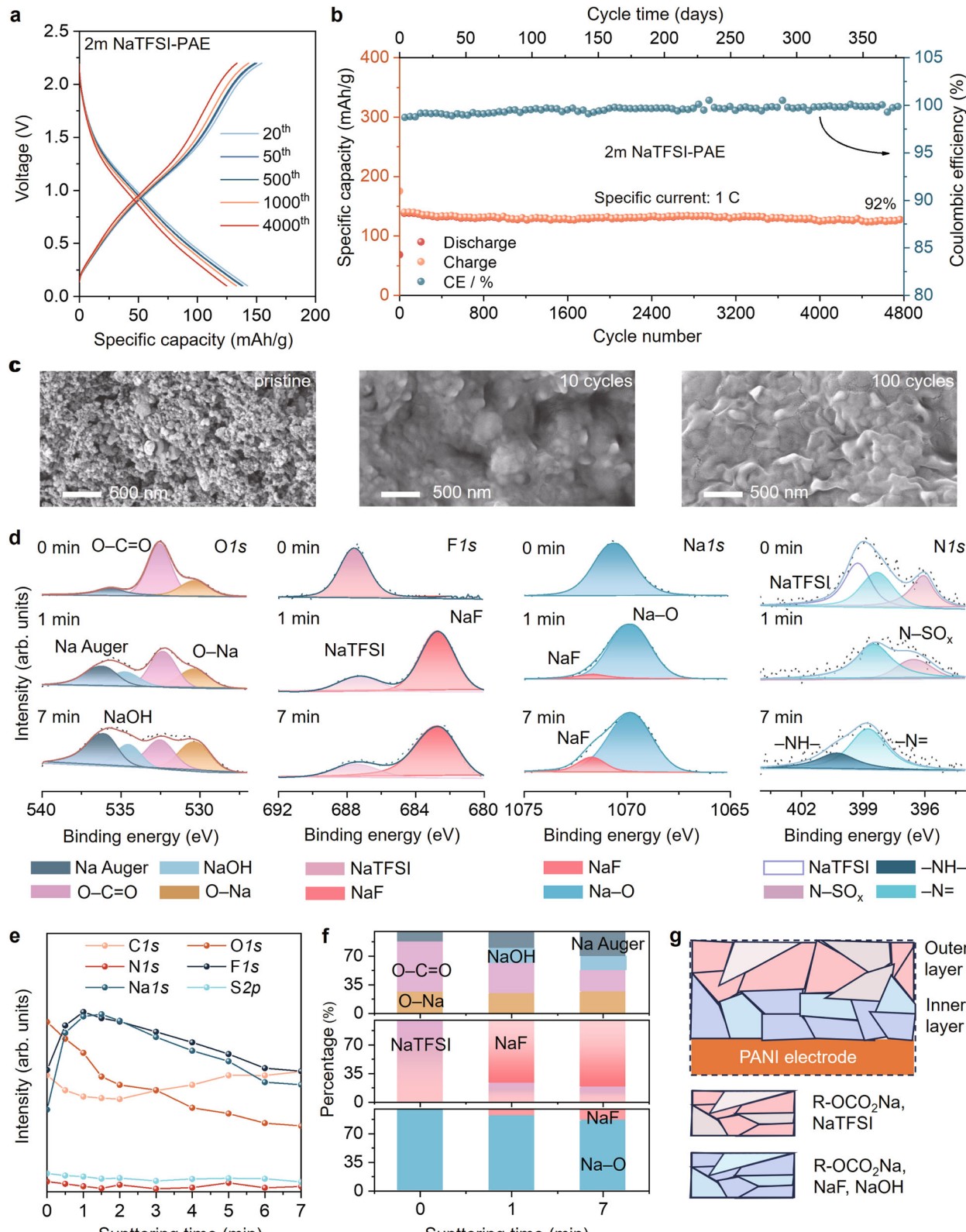

**Fig. 4 | Electrochemical performance and solid electrolyte interphase.**
**a** Galvanostatic charge-discharge curves of all-polymer batteries at 20, 50, 500, 1000, and 4000th cycle. **b** Cycling performance and coulombic efficiency in 4800 cycles (381 days). **c** SEM images of the PANI surface after different cycles (pristine, after 10 and 100 cycles). **d** XPS spectra (O1s, Na1s, F1s, N1s) of SEI on PANI anode after different sputtering times (0, 1, and 7 mins). **e** Intensities of different elements (O1s, Na1s, F1s, N1s, S2p, and C1s) of SEI film after different sputtering time. **f** Ratio of different Na salts of SEI after different sputtering times (0, 1, and 7 mins). **g** SEI structure on PANI anodes.

Fig. 22). However, an apparent electrolysis reaction of PANI could be observed just above 0.8 V in the 2 m NaTFSI-H$_2$O electrolyte despite its ESW being 1.75 V (Supplementary Fig. 23). FTIR was further applied to track the structural evolution of PANI cathodes during the charging process in these two different electrolytes (Supplementary Fig. 24). In the FTIR spectra, the bands at 1589 cm$^{-1}$ and 1500 cm$^{-1}$ of pristine PANI correspond to benzenoid ring stretching vibrations, which shift to 1574 cm$^{-1}$ and 1502 cm$^{-1}$ during the charging process due to the structure transformation from benzenoid (B) ring to quinonoid (Q) ring[53]. More distinct bands change at 1154 cm$^{-1}$ and 1128 cm$^{-1}$, corresponding to the in-plane deformation of Q = N = Q and B−NH−B, respectively. In 2 m NaTFSI-PAE, the band mentioned above changes of the benzenoid−quinonoid transformation in PANI cathodes are highly reversible after discharge. Conversely, bands relevant to the quinonoid (1589 cm$^{-1}$ and 1502 cm$^{-1}$) structure of PANI cathode in 2 m NaTFSI-H$_2$O electrolyte become much broader and stronger than that of pristine PANI even after fully discharging to 0.1 V. These results prove that reversible charge-discharge of the PANI cathode at high voltage can be achieved in 2 m NaTFSI-PAE. The stability enhancement of the highly oxidised PANI cathode can be explained by the low activity of H$_2$O in PAE, as mentioned in Fig. 2. Apart from the activity of H$_2$O, the anion species is quite an important factor in the stability of the PANI cathode. NaClO$_4$, NaOTF, and NaFSI were further investigated in PAE due to their smaller anions than NaTFSI. Both 2 m NaClO$_4$-PAE and 2 m NaOTF-PAE showed a wide electrochemical stability window compared with 2 m NaTFSI-H$_2$O (Supplementary Fig. 25), however, their ionic conductivities were lower than 2 m NaTFSI-PAE. The above electrolytes were further applied to the symmetric PANI batteries. Those batteries didn't exhibit better cycling stability than 2 m NaTFSI-PAE, possibly due to their stronger interactions with H$_2$O[54] (Supplementary Figs. 26−28).

Compared with the PANI cathode, the reversible n-doping of PANI anode in PAE remains a mystery because the negatively charged nitrenes on the pernigranilate show high basicity, which are typically easily protonated by protic solvent molecules. The electrochemical impedance spectroscopy (ESI) after different cycles was further utilized to evaluate the reaction between electrolyte and electrodes (Supplementary Fig. 29). Evident semicircles corresponding to $R_{ct}$ and $R_{SEI}$ appeared after the first cycle, and the charge transfer resistance significantly decreased, indicating the formation of continuous and dense solid-electrolyte interphase (SEI) layers[55]. By enlarging the linear sweep voltammetry (LSV) of PAE within − 2.0−0.5 V, a small peak at − 1.23 V also indicated the SEI layers originating from the reduction of electrolyte (Supplementary Fig. 30). The scanning electron microscope (SEM) demonstrated the morphology transition of the anode surface, which also indicates the formation of dense and smooth SEI layers during cycling (Fig. 4c). XPS was further conducted to figure out the chemical compositions of SEI (Fig. 4d and Supplementary Fig. 31). From the binding energy changes of C$1s$, O$1s$, and Na$1s$ in XPS of PANI anode after different cycles (Supplementary Fig. 32), it shows significant increasing of C−O (286.6 eV) and C = O (287.5 eV) peak signals in C$1s$, and C−O (531.2 eV), C = O (533.7 eV) and Na−O (529.5 eV) peak signals in O$1s$, indicating the main component of SEI includes sodium alkoxides (RCH$_2$CH$_2$ONa)[56,57]. The apparent Na−O peak signals in Na$1s$ further confirmed the formation of RCH$_2$CH$_2$ONa. For further understanding of the SEI structure, XPS of the anode surface with different ion sputtering times was conducted. With the sputtering time increasing, the intensity of O1s kept decreasing, but the intensity of Na$1s$ and F$1s$ increased within 1 min of ion sputtering and then decreased, indicating the SEI structures consisted of multiple components (Fig. 4e). Detailed analyses were conducted by fitting XPS spectra of N$1s$, O$1s$, F$1s$, and Na$1s$ with 0, 1, 7 min ion sputtering. After 7 min sputtering, the −NH− groups of PANI appeared, proving the SEI layers grew on the PANI surfaces. Besides, the fitting results

demonstrate the decrease of R-OCO$_2$Na, and the increase of NaOH, and NaF after ion sputtering (Fig. 4d). Although NaF may be generated from the decomposition of NaTFSI after 1 min of ion sputtering, the decomposed layers of NaTFSI should be removed after 1 min ion sputtering, which can be judged from the intensity change of byproduct C-SO$_x$ in S$2p$ (Supplementary Fig. 31). Then by comparing the ratio of different Na salt species in SEI layers by XPS of O$1s$, Na$1s$, and F$1s$ (Fig. 4f), we can determine the SEI structures shown in Fig. 4g, where R-OCO$_2$Na is the main component in SEI, accompanied with NaTFSI in the outer layer, NaOH and NaF in the inner layer.

The SEI formation mechanism was further studied. First, the surface of active carbon (AC) in the half−cell was analysed by XPS (Supplementary Fig. 33). The XPS results of AC didn't show a big difference from PANI, which means the PANI may not participate in the SEI formation. It is noteworthy that the intensity of Na$1s$ in the 1$^{st}$ cycle is weak, but the structural transformation of the PANI anode from benzenoid (B) ring to quinonoid (Q) is apparent in FTIR spectra, indicating that the PANI anode may be reduced by protons from the H$_2$O (Supplementary Fig. 19). The XPS results of the PANI anode after ion sputtering also revealed that NaOH was generated in the SEI layer. Hence, although the PANI anode was not directly involved in SEI components, it may induce the generation of NaOH and facilitate the formation of SEI layers. Then, the effect of anion species on the SEI structure was further analysed with the PANI anode after charge-discharge in 2 m NaClO$_4$-PAE, 2 m NaOTF-PAE, and 2m NaFSI-PAE for 10, 50, and 200 cycles (Supplementary Figs. 34−36). Apart from 2 m NaClO$_4$-PAE, the XPS results of PANI anodes are almost the same as those in 2 m NaTFSI-PAE, indicating a similar surface composition of SEI attributed to PEGDME decomposition.

The above systematic studies revealed that PEGDME plays the most significant role in SEI formation, and the main component of SEI is R-OCO$_2$Na. Hence, our PAE not only improved the high electrochemical stability of H$_2$O but also benefited SEI formation, which contributed to the cycling stability of all-polymer ASIBs.

**Processibility, flexibility and recyclability of all-polymer battery**

Thanks to the intrinsic advantages of polymeric materials in processability, flexibility and recyclability, we successfully achieved large-scale fabrication and recycling of all-polymer flexible batteries based on the above studies. Large-scale polymer electrode films were continuously fabricated via the roll-to-roll technique, where the pre-prepared slurry was roll-pressed onto the flexible conductive substrates. Then, the as-fabricated electrode films were collected after drying (Fig. 5a). The all-polymer film batteries can be efficiently made by sandwiching the membrane soaked with an electrolyte between prepared electrode films due to the simple construction of symmetric electrodes. The film batteries exhibit excellent flexibility, which can stably charge-discharge with specific capacities of 135 mAh/g (Fig. 5b and Supplementary Fig. 37), even under 90° bending (Fig. 5c). The all-polymer fibre batteries can also be integrated and prepared by coating the slurry onto the conductive fibre substrate, wrapping the separator, twisting two electrode fibres, and sealing the electrode fibres in PTFE tubes filled with electrolyte (Fig. 5d). The fibre batteries exhibit excellent flexibility, which can stably charge-discharge with specific capacities of 122 mAh/g (Fig. 5e), even under 90° bending (Fig. 5f). After 100 cycling times, the thickness of this film battery didn't show apparent change, indicating that there is no continuous electrolysis of electrolyte (Supplementary Fig. 38). Besides, the flexible batteries kept working even after cutting, proving its high safety when applied in wearable electronics (Supplementary Fig. 39).

One all-polymer film battery can output a 2.2 V terminal voltage for powering a temperature-humidity recorder (Supplementary Fig. 40). The terminal voltage and power can be multiplied by connecting several film batteries in series, for example, two connected film

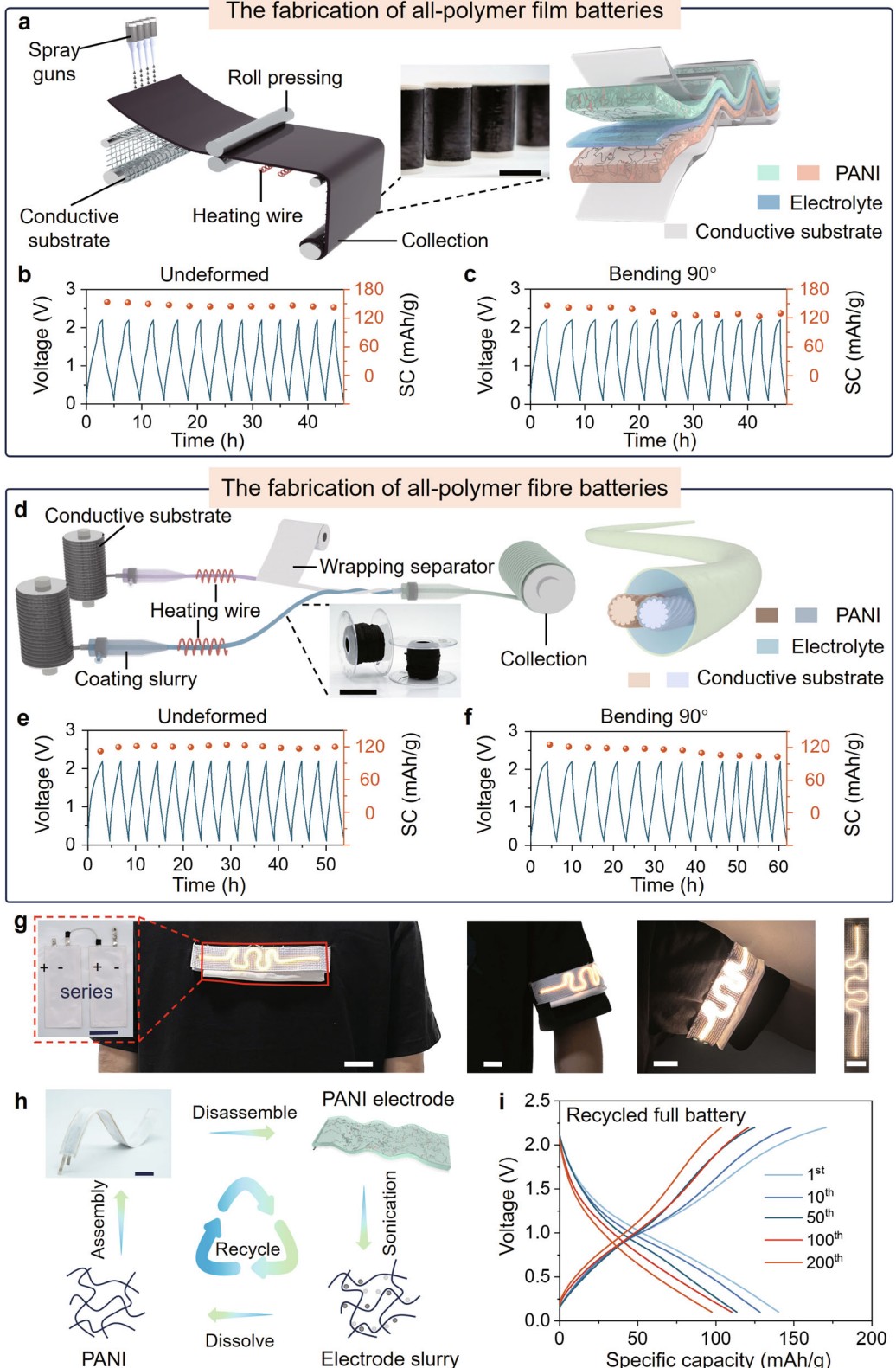

**Fig. 5 | Flexible all-polymer aqueous sodium batteries. a** Roll-to-roll scale preparation of PANI electrodes. Scale bar, 5 cm. **b**, **c** Galvanostatic voltage profiles of flexible all-polymer film batteries under different deformations. SC (mAh/g): specific capacity (mAh/g). **d** Schematic of continuous fabrication of all-polymer fibres batteries. Scale bar, 5 cm. **e**, **f** Galvanostatic voltage profiles of flexible all-polymer fibres batteries under different deformations. SC (mAh/g): specific capacity (mAh/g). **g** Photograph of a flexible LED powered by flexible films all-polymer batteries group. Scale bar, 3 cm. **h** Schematic of the recycling process of PANI from waste all-polymer ASIBs. Scale bar, 4 cm. **i** Galvanostatic voltage profiles of all-polymer ASIBs fabricated with recycled PANI at 1st, 10, 20, 50, 100 and 200th charge-discharge.

batteries can output over 3.0 V terminal voltage for power wearable light-emitting device (Fig. 5g). Although conductive polymer active materials typically lack apparent charge/discharge platform due to the changing band gap during the doping/dedoping, the charge/discharge of batteries can be controlled under the constant voltage in the practical application. By circuit design and process optimisation, the flexible all-polymer film batteries could be applied to various wearable electronics in the future.

Recycling electrode materials is essential to develop sustainable, flexible power technologies that reduce waste pollution and reuse resources. Our all-polymer ASIBs not only replace the use of metal elements in electrodes but also have the advantage of recyclable polymer electrodes to avoid potential waste pollution after large-scale application. Fortunately, recycling PANI is much easier than other inorganic electrode materials. After long-term use of the all-polymer ASIB, PANI can be separated from the electrode materials by washing with methylpyrrolidone (Fig. 5h). The structure of recycled PANI was characterised by FTIR spectra (Supplementary Fig. 41), exhibiting the typical stretching modes of $C = N$ (1650 cm$^{-1}$) and $C = C$ (1520 cm$^{-1}$). The recycled PANI can be used as electrode active material again, whose performance was characterised by GCD measurements in all-polymer batteries, demonstrating an initial capacity of 140 mAh/g at 1 C rate and 70% capacity retention after 200 cycles (Fig. 5i and Supplementary Fig. 42). The successful recycling of PANI electrodes through simple solvent processing shows that the significant advantages of an all-polymer ASIBs for sustainable energy storage technologies lie in abundant resources, easy reuse, and less waste.

## Discussion

In summary, we achieved all-polymer aqueous batteries by using intrinsic PANI as symmetric electrodes, which is considered a significant challenge due to the instability of fully oxidised pernigraniline salt and fully reduced metal pernigranilate in neutral aqueous electrolyte. The key to tackling the instability issues of PANI electrodes is using PAE with an electrochemical stability window of 3.2 V and dense solid-electrolyte interphase. Systematic characterisations, including FTIR, Raman spectra, and NMR, revealed the structure changes of H$_2$O and solvation layers of Na-ion in PAE, which led to a decrease in H$_2$O activity. DFT simulations and spectral characterisations further confirmed that PANI can function as symmetric electrodes through a dual-ion doping mechanism in PAE, which is impressive for achieving both n-doping and p-doping of PANI in an aqueous environment. Based on the PAE electrolyte, the assembled all-polymer batteries achieved breakthrough performances in ASIBs with a specific capacity of 139 mAh/g, energy density of 153 Wh/kg, 92% retention after 4800 cycles, surpassing most state-of-the-art aqueous Na-ion batteries (Supplementary Table 4). FTIR further revealed that the irreversible deprotonation of PANI cathode can be avoided in PAE even under high voltage. Then, XPS and SEM revealed the SEI formation on the PANI anode, allowing the n-doping of PANI in an aqueous electrolyte. Finally, the all-polymer battery exhibits excellent processability, flexibility, and sustainability, which is aligned with our goals to achieve large-scale application of flexible power technologies. The fabricated flexible battery still delivers a specific capacity of 135 mAh/g, which exceeds that of most advanced film/pouch Li/Na-ion aqueous batteries (Supplementary Table 5). Even batteries reassembled with recycled PANI exhibit ideal performance. Thus, high-performance and sustainable all-polymer ASIBs were achieved based on the economical conductive polymer PANI and low salt concentration PAE. We believe this work will propel the development of high-energy and low-cost organic electrodes and aqueous electrolytes, leading to sustainable, flexible energy storage independent of mineral resources.

## Methods

### Materials

Unless otherwise stated, all materials were purchased from commercial suppliers (Sigma Aldrich, Adamas, Aladdin, and Energy Chemical et al.), and used without further purification. Polytetrafluoroethylene (PTFE, 60 wt%, Canrd Co.), polyvinylidene fluoride (PVDF) binder (900, Arkema), deionized water (purified on a MilliQ device from Millipore), isopropyl alcohol (IPA, AR, Aladdin Co.), ethanol (ETH, AR, Adamas), toluene (TL, AR, Shenzhen Changtai Chemical Technology Co.), N-methyl-2-pyrrolidinone (NMP, EL, Adamas), ammonia hydroxide (NH$_3$·H$_2$O, AR, Adamas) poly(ethylene glycol) (PEGDME250/500, AR, Adamas), G1645 styrene-ethylene-butylene-styrene (SEBS, Kraton, USA), aniline (99%, RG, Adamas), ammonium persulfate (APS, 99.99% RG, Adamas), 3,4,9,10-perylenetetracarboxylic diimide (PTCDI, 98%, RG, Adamas), perylene-3,4,9,10-tetracarboxylic dianhydride (PTCDA, 98%, RG, Adamas), 1,4,5,8-naphthalenetetracarboxylic dianhydride (NTCDA, 97%, RG, Adamas), polypyrrole (PPy, macklin), sodium bis (fluorosulfonyl) imide (NaTFSI, EL, Dadao New material Co.), silver/silver chloride reference electrode (Ag/AgCl, SSCE, Tianjin Ada Hengsheng Technology Development Co.), acetylene black (AB, Kappa 100, Canrd Co.), ketjen black (KB, EC300J, Canrd Co.), titanium meshes (Ti meshes, 200 mesh, Hebei Kangwei Metal Materials Co.), stainless steel meshes (400 mesh, SS316, Hefei Wenghe Metal Materials Co. Ltd., China), Whatman glass fibre membrane (GF/D, Canrd Co.), nickel strips (Canrd Co.), CR2032 case (316 L, Canrd Co.), coin battery sealing machine (MSK-110, Shenzhen Kejing Star Technology Co.), argon (Ar, high-purity argon, Shenzhen Shente Industrial Gas Co.).

Polyaniline (PANI, emeraldine base) was synthesized through two-step reactions according to the literature[58]. The procedure was as follows: 0.93 g of aniline monomer (10 mmol) was added to 1 M HCl solution. Ammonium persulfate (APS, 10 mmol) was then slowly added at 0–5 °C. The reaction mixture was stirred for four hours. At the end of the reaction, the intermediate was obtained by filtration, washing, and drying. Next, 1.00 g of the intermediate was dispersed in 50 mL of 6 M NH$_3$·H$_2$O and stirred at room temperature for 24 h. The solid product was then separated, washed, and dried according to the previous procedure to obtain the final product. The synthesized PANI was characterised by FTIR, whose synthesis route and structure information is shown in Supplementary Fig. 1 and Supplementary Table 2.

Poly-(naphthalene four formyl ethylenediamine) (PNFE) was synthesized through one-step reactions according to the literature[17]. More specifically, equimolar amounts of NTCDA and ethylenediamine were added to the NMP solvent. After refluxing the mixture for 6 h, the precipitate was filtered and thoroughly washed several times with ethanol. The precipitate was then vacuum-dried at 120 °C for 12 h. Subsequently, the precipitate was heated under a nitrogen atmosphere at 300 °C for 8 h, yielding a brown powder. The overall yield of this synthesis was ~80%. The synthesized PNFE was characterised by FTIR, whose synthesis route and structure information are shown in Supplementary Fig. 2 and Supplementary Table 3.

### Preparation of 2 m (mol/kg) NaTFSI polymer-aqueous electrolyte (PAE)

PAE was prepared by mixing PEGDME (250/500) with deionized water in a weight ratio of 47:3, followed by dissolving 2 m NaTFSI into the PEGDME-H$_2$O cosolvent. PEGDME450 was prepared from a blend of PEGDME250 and PEGDME500 in a 1:4 weight ratio.

### Electrode preparation

Preparation of the AB working electrode and AC counter electrode. Firstly, a mixture of AB powder and PTFE was mixed well with a weight ratio of 1:1 in IPA. Subsequently, the slurry was uniformly applied onto a Ti mesh to fabricate the electrode. Finally, the as-prepared electrode was dried at 80 °C under vacuum for 12 h. The preparation of the AC

counter electrode was the same as the AB working electrode, where AC replaced the AB.

## Assembly of all-polymer battery

Fabrication of the symmetric PANI coin batteries. Firstly, the active materials (PANI), conductive agent (KB), and binder (PTFE) were mixed well with a mass ratio of 7:2:1 in IPA to form a slurry, which then was rolled into a uniform film. Next, the electrode sheet was prepared by pressing the electrode film onto the current collector (Ti mesh), followed by vacuum dying at 80 °C for 12 h. The electrode sheet was then cut into disc electrodes with a diameter of 12 mm. Asymmetric PANI coin battery was assembled with two electrode sheets, a piece of Whatman glass fibre membrane, and 2 m NaTFSI-PAE with CR2032-type coin battery sequentially in a glove box. Before testing, the as-assembled batteries were kept standing for more than 8 h.

Fabrication of film batteries. The electrode film preparation was the same as the coin cells. The flexible electrodes were obtained by pressing 2.0 cm × 6.0 cm film onto a 2.5 cm × 7.0 cm current collector (the flexible stainless-steel mesh). Nickel strips were affixed as electrode tabs to the edges of flexible electrodes. Then a glass fibre membrane separator with a thickness of ~ 200 μm was sandwiched by two flexible electrodes. Subsequently, a SEBS pouch was applied as an encapsulation for flexible batteries. After the electrolyte was injected into the SEBS pouch, the battery was sealed under vacuum. Finally, the resulting battery was encapsulated using parylene (~ 20 μm in thickness). SEBS pouch was prepared by dissolving 6.0 g G1645 SEBS into 30.0 mL toluene, along with 0.5 mL a 1.0 wt% white dye inclusion, subsequently moulding into SEBS films and sealing all edges of two SEBS films with a heat press.

Fabrication of fibre batteries. CNT fibres were selected as the current collectors to estimate the performance of fibre batteries. The carbon nanotube fibres were synthesized by a continuous chemical vapor deposition method[59]. The electrode slurries were prepared by mixing active materials (PANI), conductive agent (KB), and binder (PVDF) in the weight ratio of 7:2:1 with NMP as solvent. The solid contents of electrode slurries were around 10%. Dip-coating processes were used to apply PANI slurries onto the CNT fibres, followed by vacuum drying at 80 °C. After wrapping one fibre electrode with a separator, it was twisted with another fibre electrode. Finally, the twisted fibre electrodes were sealed in a heat-shrinkable tube filled with electrolytes and stood overnight for electrolyte infiltration.

## Electrochemical testing

The electrochemical stability windows of electrolytes were evaluated using a three-electrode device with an AB working electrode (~ 0.5 mg/cm$^2$), an AC counter electrode (15–20 mg/cm$^2$), and the SSCE reference electrode. The test was conducted in an argon (Ar) atmosphere. The electrochemical performances of all-polymer batteries were evaluated with coin cells. The mass loading of the electrodes was between 1.5–2.6 mg/cm$^2$, 70 wt% of which is active material. The diameter of disc electrodes is 12 mm. The diameter and thickness of separators are 20 mm and 200 μm respectively. All assembly and disassembly processes of batteries were performed in a dry Ar-filled glove box ($H_2O$ < 0.1 ppm, $O_2$ < 0.1 ppm Mikrouna). Cyclic voltammograms (CVs) were tested on a CHI instrument electrochemical workstation (Corrtest CS2350M) at a scan rate of 0.1 mV/s between 0.1 and 2.2 V. The electrochemical impedance spectroscopy (EIS) tests were measured by a CHI instrument electrochemical workstation in the frequency range of $10^{-2}$–$10^{5}$ Hz at the amplitude of 5 mV. The galvanostatic cycling test was conducted on NEWARE battery testing system (CT-4008T-5V50mA, Shenzhen, China)), and all batteries were pre-cycled between 0.1–2.2 V with 1 C specific current for 10 cycles before cycling measurements (Supplementary Fig. 43). Cycling experiments were performed between 0.1–2.2 V with a specific current of 147 mA/g in an environmental chamber with 28 °C.

## Ionic conductivity test

Before testing, Ar gas was introduced for 15 min to ensure the test environment was free from oxygen interference. A three-electrode system was used for AC impedance testing, with a frequency range of $10^{-2}$–$10^6$ Hz and an amplitude of 5 mV. The AB electrode served as the working electrode, the AC electrode functioned as the counter electrode, and the SSCE acted as the reference electrode. The ionic conductivity was calculated using the formula:

$$\sigma = \frac{d}{R_b \bullet A} (\text{S/cm}) \qquad (3)$$

$\sigma$: Ionic conductivity, $d$: the thickness of electrolyte, $R_b$: Bulk resistance of the electrolyte, $A$: Electrode area

## Material characterisations

The nuclear magnetic resonance (NMR) spectra were obtained from a BRUKER AVANCE III 400 MHz NMR Instrument (in DMSO-$d_6$ at room temperature). The Fourier transform infrared spectrometer (FTIR) was recorded using KBr pellets or ATR modes on a Thermo Nicolet 6700 with a wavenumber range of 4000 to 400 cm$^{-1}$. Gemini SEM360 scanning electron microscope was used to characterise morphology.

## Computational details

Spin-polarised density function theory (DFT) calculations were performed using the Vienna Ab initio Simulation Package (VASP). The projector augmented wave (PAW) method was used to describe the interactions between ion cores and valence electrons. The Perdew-Burke-Ernzerhof (PBE) functional within generalised gradient approximation (GGA) was utilized to describe the exchange-correlation energy. For structure relaxations, the Brillouin zone was sampled by the Monkhorst-Pack (MP) scheme and 2 × 1 × 1 k points. The kinetic energy cutoff for the plane wave basis set was chosen to be 520 eV. The force on each atom in the structural optimisation was set as $2 \times 10^{-2}$ eV/Å, and the electronic energy convergence criterion was $1 \times 10^{-6}$ eV. The van der Waals interactions between the ions and PANI electrodes were considered using the DFT-D2 method.

The theoretical specific capacity ($C_{\text{theoretical}}$) of the dual-ion doping process is calculated using the formula[60]:

$$C_{\text{theoretical}} = \frac{nF}{3.6\,M} (\text{mAh/g}) \qquad (4)$$

Where $n$ is the number of electrons in the dual-ion doping process, $F$ is the Faraday constant, and M is the mole weight of the reactant (g/mol).

The specific energy (Wh/kg) of the coin-type battery and film battery were calculated by the following formula:

$$E_g = \frac{E_D}{m_e} \qquad (5)$$

where $E_g$ is the specific energy (Wh/kg) of the battery, $E_D$ is the discharging energy (Wh) of the battery provided by the data processing software of the NEWARE battery testing system, and $m_e$ is the loading weight (kg) of the active material in the cathode.

## Electrode active materials recycling

Firstly, the battery was disassembled, and the cathode and anode were separated. They were soaked in a solvent mixture of deionized water and ethanol (1:1 in volume ratio) for 2 h, followed by three times washing with deionized water to remove residual electrolytes. Subsequently, they were subjected to 4 h of ultrasonication in IPA solvent to separate the electrode materials from the stainless-steel mesh current collectors, giving a slurry residue. The slurry residue was then

dissolved in NMP by stirring for 12 h, filtered, and the filtrate was collected. Finally, the collected material was dried using a rotary evaporator to obtain the recycled product.

### FTIR, Raman, and XPS spectroscopy of electrode

In an Ar-filled glove box, the electrode sheets were collected from the batteries at various charging and discharging stages, followed by washing with deionized water and ethanol to eliminate any remaining electrolytes. Subsequently, they were vacuum-dried at 70 °C for 8 h and stored in an Ar atmosphere. Before each test, the specific electrodes were swiftly removed from the Ar-filled environment.

## Data availability

All data needed to evaluate the conclusions in the paper are present in the paper and/or the Supplementary Materials. Source data are provided in this paper.

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

## Acknowledgements

This work was financially supported by the National Natural Science Foundation of China (52103300), Guangdong Basic and Applied Basic Research Foundation (2023A1515010572), Shenzhen Science and Technology Programme (JCYJ20210324132806017, GXWD20220811163904001). The authors thank the staff from the Shanghai Synchrotron Radiation Facility (SSRF) at BL02U2. We thank Lei Chen (East China University of Science and Technology) for equipment assistance.

## Author contributions

X.F. and S.H. conceived the idea and supervised the project. Y.H. and K.J. designed the experiments and conducted characterisations, electrochemical measurements, and analyses. K.J., J.J., Q.G. and J.C. conducted characterisations of materials. Z.L. and Y.Z. conducted DFT computations. Q.L. drew schematic diagrams. H.S. performed Raman characterisation. X.F., S.H., Y.H., K.J., X.D., D.Z. and R.L. discussed the results and participated in preparing the paper. Y.H. and K.J. contributed equally to this work. All authors agreed on the manuscript and approved the submission.

## Competing interests

The authors declare no competing interests.
