## [Transparent Peer Review file · Nature Communications]

Energetic and durable all-polymer aqueous battery for sustainable, flexible power

Corresponding Author: Professor Xiulin Fan

Version 0:

Reviewer comments:

Reviewer #1

(Remarks to the Author)

Yang Hong et al described the fabrication of all-PANI aqueous sodium-ion battery using a polymer-aqueous electrolyte designed to stabilize electrode redox products by modulating the solvation layers and forming a solid-electrolyte interphase. The rationale of using PANI as the electrode materials in the introduction part could be refined further, since PANI also has significant drawbacks (e.g., sloppy voltage profiles (which also observed in the current work), relatively low-degree of doping level) together with the advantages mentioned in the introduction. One of the major novelty of this work is using polymer-based aqueous electrolyte to enable the stable cycling of PANI anode by the formation of SEI. However, the current work still need more detailed analysis of the SEI at anode. Therefore, given the good novelty but less thorough investigation of this work, the reviewer suggests publication of this work after the questions raised are resolved.

1. Please explain why avoid using high-concentration sodium-ion electrolytes is preferred, since Na is very abundant.
2. It should be "H-O stretching" in Fig. 2c, not "H-O bending".
3. Why the FTIR intensity of PAE is significantly lower than 2m NaTFSI?
4. How does the Na⁺ solvation region look like in Raman spectra? Raman spectroscopy is a common method to evaluate the solvation of cations in electrolytes. In scheme 1c, it indicates that Na⁺ only solvated by PEGDME, does this supported by Raman analysis?
5. Could the authors explain why breaking the hydrogen bonding between H₂O molecules (and forming weaker H-bonding with PEGDME or becoming completely isolated) is beneficial in terms of enlarging the ESW? Normally, the activity of water can be decreased by the coordination to cations, such as in water-in-salt electrolytes where almost all H₂O molecules bind strongly to cations thus reducing the activity of H₂O. Therefore, why does the removal of hydrogen bonding have similar effects?
6. According to Nat. Mater. 2013, a b value between 0.7 and 0.8 can arise from numerous sources including an increase of the ohmic contribution (active material resistance, solid-electrolyte interphase resistance) and/or diffusion constraints/limitations. Therefore, it's not quite appropriate to say that b=0.75 proves that the PANI cell performs like a battery. A more detailed analysis using $i(v) = k_1u + k_2v^{1/2}$ is recommended (see <https://www.nature.com/articles/s41578-019-0142-z>).
7. According to Supplementary Fig. 11b and 11d, the IR drop during GCPL cycling is quite large, could the authors provide an explanation for the reason? Is this due to electrolyte resistance, electrode resistance, or both?
8. On the PANI cathode side, the doping and de-doping of anions determine the performance and stability. Therefore, the reviewer was curious if the authors have tested anions other than TFSI-? TFSI anion is quite large compared with ClO₄- or sulfate, will using smaller anions further improve the stability of PANI without sacrificing the performance?
9. Please include the actual FTIR spectra measured during cycling (corresponding to Fig. 3c,e) in the supplementary materials.
10. Please indicate which mass are the specific capacity and energy density of all-PANI battery based on. Are they based on the total mass of active material in the cathode+anode? Or just based on one electrode? Similarly, please indicate in the method part that how were the specific capacity values calculated. Are they based on active material mass or the mass of the total electrode (AM+carbon+binder)?
11. Supplementary Fig. 14 exhibits a strong anodic peak at ~1.8 V in the first cycle but not in the subsequent cycles. Could the authors explain?
12. The authors used XPS to characterize SEI on PANI anode after cycling. But the reviewer is curious if the authors have conducted depth profiling of the cycled PANI anode to confirm there is actually SEI on the surface? Basically, the XPS spectra should become similar to pristine PANI after ion sputtering which removes the surface (e.g., Na signal should be

gone,).

13. Please provide more detailed discussion of the structure/component of SEI, including a proposed structural scheme of SEI and formation mechanism. For example, does the SEI form due to the decomposition of PANI or polymer electrolyte or salt? Does different Na salt affect the formation of SEI? Since enabling the PANI anode is one of the major novelty of this work, a more detailed analysis is needed.

14. EIS studies of the PANI anode were only conducted at the cycled state (charged to 2.2 V), but not at a discharged state. Please provide the EIS data when the cell is discharged to 0.1 V (or close to 0.1 V) after different cycles and corresponding interpretation.

Reviewer #2

(Remarks to the Author)

This paper reports a flexible aqueous sodium-ion battery (ASIB) featuring symmetric polyaniline (PANI) electrodes, and polyethylene glycol dimethyl ether (PEGDME) electrolyte to modulate the solvation layers and form a solid-electrolyte interphase. The assembled symmetric PANI ASIBs demonstrated a high capacity (139 mAh/g) and a stable and long cycle life (>4800 cycles with 92% capacity retention). Overall, the characterization of all-polymer aqueous batteries are systematically investigated and the results look good. I think it should be published after minor revisions, as listed below.

- 1) The manuscript claims that the H-O bond in the water is loose by adding PEGDME, reducing the activity of H₂O in the electrolyte. It seems the ether groups may play an essential role in improving the electrochemical stability of electrolytes. The manuscript only tested a single material in this study. It is unclear whether the same conclusions will remain plausible with other ether-rich polymers.
- 2) The manuscript tested low concentrations of salt only in this study. It would be better to see the concentration dependence of electrochemical stability as a high concentration is predicted to enhance the ionic conductivity.
- 3) In Figure 4a, the capacity decreases significantly over the cycles, which is contrary to the results in Figure 4b. Please double check the data used.
- 4) In line 182, it mentioned that the peak of S-N-S (879 cm⁻¹) periodically occurred in the figure.3c. But the range of wavenumber in fig.3c is from 1000 to 1800 cm⁻¹.
- 5) In fig.3f, please explain why the difference between intensities of Na 1s from two discharging cycles is relatively large.
- 6) In fig.4a, although the all-PANI battery exhibited a high specific capacity at 1C, the voltage increased/decreased fast during charging/discharging process. Normally, a stable charge/discharge platform can ensure a constant and safe power supply for devices in practical applications. Could the author explain why there is no apparent platform during process?
- 7) In Page 6, it's mentioned that the SEI layers in ASIB originated from the reduction of electrolyte. As the author claimed that PANI was firstly used as anode, is it also contributed to the formation of robust SEI?

Reviewer #3

(Remarks to the Author)

I co-reviewed this manuscript with one of the reviewers who provided the listed reports. This is part of the Nature Communications initiative to facilitate training in peer review and to provide appropriate recognition for Early Career Researchers who co-review manuscripts

Version 1:

Reviewer comments:

Reviewer #1

(Remarks to the Author)

The additional experimental results and detailed explanations/answers have resolved all my questions. Therefore, I recommend this manuscript for publication without further revision.

Reviewer #2

(Remarks to the Author)

The authors have satisfactorily addressed all the comments from the reviewer and no further revision is needed.

Reviewer #3

(Remarks to the Author)

Summary of Response to the Reviewers

(*Black italic: Reviewer's comments*; Black type: Our response; Blue type: Key revisions)

We appreciate constructive comments and suggestions for improving the manuscript. We have acted on them as documented point-to-point below. To summarize:

Experimentally:

1. We carried out a series of new experiments and analyses, including sodium nuclear magnetic resonance spectroscopy ($^{23}\text{Na-NMR}$), ether peaks fitting of Raman spectra on 2m NaTFSI-PAE electrolyte with control groups of on 2m NaTFSI-H₂O, 2m NaTFSI-PEGDME to improve the structural studies of solvation layers in our electrolyte.
2. We carried out a series of new experiments and analyses, including linear sweep voltammetry (LSV), and ionic conductivities measurement of NaTFSI-PAE with different concentrations (2m, 3m, 4m, and 5m NaTFSI).
3. We carried out a series of new experiments, including new Ar⁺ sputtering-assisted X-ray photoelectron spectroscopy (XPS) on PANI anode with a control sample of active carbon after cycles to improve the structural studies of solid electrolyte interphase (SEI) of our batteries.
4. We carried out a series of new experiments, including LSV, ionic conductivities measurement, cyclic voltammograms (CVs), galvanostatic charge-discharge cycling, SEI XPS analysis of symmetric PANI batteries with different sodium salts to study the effect of anion species on electrolyte properties, battery performances, and SEI structure.

We have revised 3 **Figures (Figures 2-4)** in the revised Manuscript, **22 Figures (Supplementary Figures 9, 10, 13, 14, 16, 17, 19-21, 24-29, 31-36, 44)** in the revised Supplementary Information, and revised paragraphs to address the Reviewers' comments adequately.

A point-to-point response to comments is provided below:

Response to Reviewer #1:

Comment #0:

Yang Hong et al described the fabrication of all-PANI aqueous sodium-ion battery using a polymer-aqueous electrolyte designed to stabilize electrode redox products by modulating the solvation layers and forming a solid-electrolyte interphase. The rationale of using PANI as the electrode materials in the introduction part could be refined further, since PANI also has significant drawbacks (e.g., sloppy voltage profiles (which also observed in the current work), relatively low-degree of doping level) together with the advantages mentioned in the introduction. One of the major novelties of this work is using polymer-based aqueous electrolyte to enable the stable cycling of PANI anode by the formation of SEI. However, the current work still needs more detailed analysis of the SEI at anode. Therefore, given the good novelty but less thorough investigation of this work, the reviewer suggests publication of this work after the questions raised are resolved.

Response: Thank you very much for the important comments and inspiring suggestions. We have conducted a suite of experiments to address all the reviewer's concerns. For convenience, the main revisions are discussed in the following point-to-point answers to the reviewer's questions and all main revisions are marked with **blue font** in the revised Manuscript. To explain the rationale of using PANI as the electrode materials, we added more descriptions about the advantages and disadvantages of PANI in the introduction part of the Manuscript as follows:

(Lines 69-70, Paragraph 3 of Page 2 in the revised Manuscript)

“Besides, the simple chemical structure of PANI also provides us the most typical example for understanding the working mechanism of all-polymer aqueous batteries.”

(Lines 73-74, Paragraph 3 of Page 2 in the revised Manuscript)

“Hence, almost all aqueous devices based on symmetric PANI electrodes are supercapacitors without evident redox voltage and efficient doping.”

(Lines 78-80, Paragraph 4 of Page 2 in the revised Manuscript)

“PANI and carbonyl derivatives are the most well-studied cathode and anode materials in aqueous batteries. Initially, we assembled a series of ASIBs based on PANI and carbonyl derivatives as a pre-experiment for developing all-polymer ASIBs (**Supplementary Table 1**).”

Comment #1:

Please explain why avoid using high-concentration sodium-ion electrolytes is preferred, since Na is very abundant.

Response: Thank you for your suggestion! High-concentration sodium-ion electrolytes are also a good candidate for broadening the electrochemical stability window and forming SEI on the anode. However, considering the higher mass density and potential corrosivity of high-concentration sodium-ion electrolytes, PEGDME possesses lower density and better biocompatibility than NaTFSI when applied in wearable electronics. Besides, although Na is very abundant, the cost of NaTFSI is still higher than PEGDME, and high concentration F-containing electrolytes will bring potential environmental risk.

Considering the reviewer's comments, we have provided a more detailed discussion about the advantages of PAE electrolytes compared with high-concentration sodium-ion electrolytes in the revised Manuscript. In addition, we also tried to increase the concentration of NaTFSI in PAE, but the result shows that higher concentrations will lead to lower ionic conductivities due to the high viscosity. The related discussions and **new Supplementary Fig. 9** have been added in the revised Manuscript as follows:

(Lines 88-93, Paragraph 1 of Page 3 in the revised Manuscript)

“High salt concentration is an efficient method for improving the stability of aqueous electrolytes, but it may bring concerns about high weight and potential corrosivity when applied in wearable electronics. To satisfy the requirements of flexibility, portability, and sustainability of flexible batteries, we chose polyethylene glycol dimethyl ether (PEGDME) as the modulator for controlling the activities and dynamics of H₂O molecules due to its advantages of low cost, low density, low volatility, and high biocompatibility.”

(Lines 124-128, Paragraph 3 of Page 3 in the revised Manuscript)

“Besides, the salt concentration dependencies of the ionic conductivities and the ESW of PAE were also studied (**Supplementary Fig. 9**). With the salt concentration increasing from 2m to 5m, the ionic conductivities decreased due to the sharply increasing viscosity. The ESW of 4m NaTFSI-PAE also didn't show an obvious increment compared 2m NaTFSI-PAE.”

new Supplementary Fig. 9. a, Ionic conductivities of 2m, 3m, 4m, and 5m NaTFSI-PAE. **b**, Electrochemical stabilities of 2m, and 4m NaTFSI-PAE.

Comment #2:

It should be “H-O stretching” in Fig. 2c, not “H-O bending”.

Response: Many thanks for your kind reminder! We have revised this description in **Fig. 2c**. Besides, **Fig. 2c** and **Fig. 2d** in the old version have been merged into **new Fig. 2c** in the new version to make the Fourier-transform infrared spectroscopy (FTIR) clearer.

The related figures have been revised in Manuscript as follows:

(Figure 2c in the revised Manuscript)

new Fig. 2 | Electrolyte design and hydration structure. c, FTIR spectra of H₂O, 2m NaTFSI-H₂O, and 2m NaTFSI-PAE.

Comment #3:

Why the FTIR intensity of PAE is significantly lower than 2m NaTFSI?

Response: The higher FTIR intensity of 2m NaTFSI than PAE is due to the higher water content in 2m NaTFSI corresponding to 3300-3500 cm⁻¹ and 1638 cm⁻¹. The PEGDME concentration of PAE is around 94 wt%. To avoid such confusion, we revised the FTIR spectrum with the wavenumber range from 1000 to 4000 cm⁻¹ to show the FTIR signal of PEGDME in **new Fig. 2c**. The revised figures have been shown in **Comment #2**.

Comment #4:

How does the Na⁺ solvation region look like in Raman spectra? Raman spectroscopy is a common method to evaluate the solvation of cations in electrolytes. In scheme 1c, it indicates that Na⁺ only solvated by PEGDME, does this supported by Raman analysis?

Response: We appreciate this significant question, which helps us think deeper and study the Na⁺ solvation region. Generally, the ether group has a stronger coordination ability than H₂O, but whether H₂O participates in the solvation layers is uncertain. To directly understand the differences in the Na⁺ solvation region, we added a series of experiments, including ²³Na NMR of 2m NaTFSI-H₂O, 2m NaTFSI-PAE, and 2m NaTFSI-PEGDME, as well as FTIR and Raman analysis of PEGDME. From the ²³Na NMR results, we can see a great change in the chemical shift of ²³Na in different electrolytes. The chemical shift of ²³Na in 2m NaTFSI-PAE (-7.51 ppm) is much closer to 2m NaTFSI-PEGDME (-9.87 ppm) rather than in 2m NaTFSI-H₂O (-0.84 ppm), which proves the main component of the Na⁺ solvation layer is PEGDME. However, the deshielding of ²³Na in 2m NaTFSI-PAE compared with 2m NaTFSI-PEGDME indicates that H₂O is also involved in the solvation layers of Na⁺ in the PAE. Besides, we analyzed the Raman spectrum of PEGDME, 2m NaTFSI-PAE, and 2m NaTFSI-PEGDME within the wavenumber range of 750-900 cm⁻¹. The solvation of Na⁺ with the ether group can also be proved by the red shift of C-O-C stretching in 2m NaTFSI-PAE when compared with PEGDME. The ratio of ether groups in Na⁺ solvation layers can be calculated based on the area of the fitted peaks in the Raman spectra. The results also prove that PEGDME is the main component in the solvation layers, but H₂O also participates in solvation.

To clarify the explanation, we reprepared **Fig. 2**, and added the figures **new Supplementary Fig. 13 and Fig. 14** by comparing the solvation structure of 2m NaTFSI-H₂O, 2m NaTFSI-PAE, and

2m NaTFSI-PEGDME.

The related discussions have been added in the revised Manuscript as follows:

(Lines 164-181, Paragraph 2 of Page 4 in the revised Manuscript)

“The solvation layer structures of Na⁺ were further studied by sodium nuclear magnetic resonance spectroscopy (²³Na NMR), and Raman spectra. The ²³Na NMR spectra show that the chemical shifts of ²³Na in 2m NaTFSI-H₂O, and 2m NaTFSI-PEGDME are around -0.84 ppm and -9.87 ppm, respectively. The ²³Na shift of 2m NaTFSI-PAE is -7.51 ppm, which is quite near 2m NaTFSI-PEGDME but far away from 2m NaTFSI-H₂O, indicating the main component of Na⁺ solvation layer is PEGDME. However, the slight movement of ²³Na in 2m NaTFSI-PAE to the downfield compared with 2m NaTFSI-PEGDME, such deshielding effect illustrates that H₂O is also involved in the solvation of Na⁺ in 2m NaTFSI-PAE. The solvation of Na⁺ with the ether group also can be proved by the red shift of C-O-C stretching in 2m NaTFSI-PAE when compared with PEGDME. Then, we evaluated the ratio of ether groups in Na⁺ solvation layers in 2m NaTFSI-PAE by calculating the percentage of ether groups in different chemical environments based on the area of fitted peaks of Raman spectra (**Fig. 2i and Supplementary Fig. 13**). Taking the solvated ether ratio in 2m NaTFSI-PEGDME as the reference, the fitting results also clearly prove that PEGDME is the main component of the Na⁺ solvation layer in 2m NaTFSI-PAE. The slightly decreased ratio of solvated ether also proves that there are small amounts of H₂O in the Na⁺ solvation layers, which can efficiently improve the ionic conductivity of 2m NaTFSI-PAE compared with 2m NaTFSI-PEGDME (**Supplementary Fig. 14**).”

new Fig. 2 | Electrolyte design and hydration structure. **d**, Raman spectra of H₂O, 2m NaTFSI-H₂O and 2m NaTFSI-PAE. **e**, Raman peak fittings of the O–H stretching vibrations with different hydrogen-bond environments, including strong, weak and non-hydrogen bonds. **f**, Percentage of fitted area for H₂O with the symmetric H-bonded (S), asymmetric H-bonded (AS), and non H-bonded (N) H₂O with increasing PEGDME concentration in 2m NaTFSI-PAE. **g**, Sodium NMR spectra of 2m NaTFSI-PEGDME, 2m NaTFSI-PAE, and 2m NaTFSI-H₂O. **e**, Raman peak fittings of the C–O–C stretching vibrations within/without solvation layers. **j**, Percentage of fitted area for other groups within/without solvation layers.

new Supplementary Fig. 13. **a**, Raman spectra of pure PEGDME, 2m NaTFSI-PEGDME, 2m NaTFSI-PAE, 2m NaTFSI-H₂O, and H₂O. **b**, Fitting of Raman spectra of PEGDME, 2m NaTFSI-PEGDME, and 2m NaTFSI-PAE. **c**, **d** Ether ratio of PEGDME, 2m NaTFSI-PEGDME, and 2m NaTFSI-PAE solvation layers.

new Supplementary Fig. 14. Ionic conductivities comparison of 2m NaTFSI-H₂O, 2m NaTFSI-PAE (2m NaTFSI-94%PEGDME-6%H₂O), and 2m NaTFSI-PEGDME (2m NaTFSI-100%PEGDME).

Comment #5:

Could the authors explain why breaking the hydrogen bonding between H₂O molecules (and forming weaker H-bonding with PEGDME or becoming completely isolated) is beneficial in terms of enlarging the ESW? Normally, the activity of water can be decreased by the coordination to cations, such as in water-in-salt electrolytes where almost all H₂O molecules bind strongly to cations thus reducing the activity of H₂O. Therefore, why does the removal of hydrogen bonding have similar effects?

Response: In light of your suggestion, we revised the description of weak H-bonding between H₂O and PEGDME. Actually, the H-bonding between H₂O and PEGDME is stronger than the H-bonding network in H₂O clusters, which should be called asymmetric H-bonding. We have revised that in the main text.

About the reason why breaking the hydrogen bonding between H₂O molecules is beneficial in terms of enlarging the ESW, enlarging ESW is considered to be contributed by the stronger H-bonding with PEGDME of asymmetric H-bonding H₂O, and the low dielectric constant of free H₂O. As the Raman fitting results of 2m NaTFSI-H₂O and 2m NaTFSI-PAE with different PEGDME concentrations shown in new **Fig. 2f**, the symmetric H-bonding H₂O molecules decreased while the non H-bonding H₂O molecules increased with increasing PEGDME concentration. The asymmetric H-bond is always dominated in 2m NaTFSI-PAE. The above results prove that H₂O molecules tend to form stronger asymmetric H-bonding with PEGDME in 2m NaTFSI-PAE, which makes H in H₂O more difficult to be reduced. Besides, compared with H-bonding water, the non H-bonding H₂O shows lower dielectric constants and polarities, which is much difficult to be electrolyzed. Then the activity of H₂O is decreased and the electrochemical stability of PAE is enhanced.

To clarify the explanation, we have reprepared **Fig. 2f** in the revised Manuscript and revised the following discussions:

(Lines 156-164, Paragraph 2 of Page 4 in the revised Manuscript)

*“In **Fig. 2f**, the percentages of the asymmetric H-bonded and non H-bonded H₂O molecules increased with increasing PEGDME concentrations, indicating the breaking of H-bond network of bulk water and the generation of free water^{48,49}. Compared with H-bonding bulk H₂O, the non H-bonding free H₂O shows lower dielectric constants and polarities, which is more difficult to electrolyze⁵⁰. Besides, it is apparent that in PAE electrolyte, the asymmetric H-bonded H₂O molecules become the dominant component (53%–69%) compared with 2m NaTFSI, which further proves that most H₂O molecules are forming asymmetric H-bonded with PEGDME. All the above characterization results demonstrate the strength enhancement of H–O bonds and the*

reduced activity of H₂O in PAE electrolyte³⁸.”

new Fig. 2 | Electrolyte design and hydration structure. d, Raman spectra of H₂O, 2m NaTFSI-H₂O, and 2m NaTFSI-PAE. **e**, Raman peak fittings of the O–H stretching vibrations with different hydrogen-bond environments, including strong, weak and non-hydrogen bonds. **f**, Percentage of fitted area for H₂O with the symmetric H-bonded (S) asymmetric H-bonded (AS), and non-H-bonded (N) H₂O with increasing PEGDME concentration in 2m NaTFSI-PAE.

Comment #6:

According to *Nat. Mater.* 2013, a b value between 0.7 and 0.8 can arise from numerous sources including an increase of the ohmic contribution (active material resistance, solid–electrolyte interphase resistance) and/or diffusion constraints/limitations. Therefore, it’s not quite appropriate to say that $b=0.75$ proves that the PANI cell performs like a battery. A more detailed analysis using $i(v) = k_1v + k_2v^{1/2}$ is recommended (see <https://www.nature.com/articles/s41578-019-0142-z>).

Response: Considering the reviewer’s comments, we have calculated the ratios of diffusion and capacitance contributions in the all-PANI cells based on your suggestions. The results show that diffusion behavior is dominant during our measurements in all-PANI cells.

To clarify the explanation, we have repared **new Supplementary Fig. 16** and revised the following discussions:

(Lines 215-225, Paragraph 2 of Page 5 in the revised Manuscript)

“For further determining the working mechanisms, diffusion/capacitance contributions at different scan rates are calculated with another empirical equation⁵²:

$$I(v) = k_1v + k_2 v^{0.5}$$

where $I(v)$ is peak current, v is potential sweep rate, k_1 and k_2 are fitting parameters. The k_1v part relies on surface absorption on the electrodes, indicating the capacitance contribution and the $k_2v^{0.5}$ part is controlled by ion diffusion inside electrodes, indicating the battery storage behavior. The ratio of diffusion and capacitance contributions in this all-PANI aqueous device under different scan rates was calculated (Supplementary Fig. 16c). The results show that when the scan rate is below 1 mV/s, diffusion behavior is always dominant. Hence, we can achieve all-PANI batteries instead of general capacitors, boosting the research interest in polymer-based high-energy aqueous devices.”

new Supplementary Fig. 16. a, CV curves at different sweep rates (v) and **b**, corresponding $\log i_p$ versus $\log v$ of the PANI electrode (i_p = peak current). **c**, Ratio of diffusion and capacitance contributions at different scan rates.

Comment #7:

According to Supplementary Fig. 11b and 11d, the IR drop during GCPL cycling is quite large, could the authors provide an explanation for the reason? Is this due to electrolyte resistance, electrode resistance, or both?

Response: The IR drop during GCPL cycling originates from electrolyte resistance due to the large distance (15 mm, Fig. R1a) between electrodes in the three-electrodes measurement system. Considering the reviewer's comments, we remeasured GCPL cycling with a new three-electrodes measurement cell, where the distance between the working electrode and counter electrode is well controlled at around 4 mm (Fig. R1b). The corresponding EIS spectra of different three-electrodes systems have been shown in Fig. R1c.

Fig. R1. **a**, Old three-electrodes measurement system. **b**, New three-electrodes measurement system. **c**, EIS spectra of different three-electrodes systems.

The GCPL cycling results have been added as new Supplementary Fig. 15.
 (*Supplementary Fig. 15 in the revised Manuscript*)

new Supplementary Fig. 15. Electrochemical behavior of the PANI electrode in 2m NaTFSI polymer-aqueous electrolyte. **a**, and **b**, CV test and galvanostatic charge/discharge in 2m NaTFSI polymer-aqueous electrolyte at 0-1 V. **c**, and **d**, CV test and galvanostatic charge/discharge in 2m NaTFSI polymer-aqueous electrolyte at -1-0 V.

Comment #8:

On the PANI cathode side, the doping and de-doping of anions determine the performance and stability. Therefore, the reviewer was curious if the authors have tested anions other than TFSI-? TFSI anion is quite large compared with ClO_4^- or sulfate, will using smaller anions further improve the stability of PANI without sacrificing the performance?

Response: Anion species are indeed quite important factors for the stability of cathode. As suggested, we tried other anion species, including OTF^- , FSI^- , ClO_4^- , and SO_4^{2-} . Unfortunately, the solubility of Na_2SO_4 in PAE was not good, as shown in **Fig. R2**, so the preparation of 2m Na_2SO_4 -PAE was failed

Fig. R2. Na₂SO₄-PAE solutions with different concentrations.

We studied the effect of anions on the electrochemical stability window and ionic conductivity based on the NaClO₄, NaOTf, NaFSI, and NaTFSI (new Supplementary Fig. 24). The results show that those salts in PAE showed suppression of the HER of electrolyte, but the OER in 2m NaFSI-PAE is still serious. Although the electrochemical stability windows of 2m NaFSI-PAE and 2m NaOTf-PAE are similar 2m NaTFSI-PAE, the ionic conductivity of 2m NaTFSI-PAE is better than other electrolytes. We further studied the electrochemical performance of symmetric PANI batteries with 2m NaClO₄-PAE, 2m NaOTf-PAE, and 2m NaFSI-PAE electrolytes. In 2m NaOTf-PAE, the battery didn't show an apparent redox peak and could not deliver high capacity. In the 2m NaClO₄-PAE, and 2m NaFSI-PAE, batteries can deliver an initial capacity like the case of 2m NaTFSI-PAE, but their coulombic efficiencies and capacity retention are too low compared with batteries using 2m NaTFSI-PAE. Although small anions seem to dope PANI cathode easily, they will form strong interactions with H₂O due to high charge density, which may cause excessive volumetric changes and are not beneficial for stability.

Considering the reviewer's comments, we have provided new Supplementary Fig. 25 and 26-28 and revised the following discussions:

(Lines 277-284, Paragraph 1 of Page 7 in the revised Manuscript)

“Apart from the activity of H₂O, the anion species is quite an important factor in the stability of PANI cathode. NaClO₄, NaOTf, and NaFSI were further investigated in PAE due to their smaller anions than NaTFSI. Both 2m NaClO₄-PAE and 2m NaOTf-PAE showed a wide electrochemical stability window compared with 2m NaTFSI-H₂O (Supplementary Fig. 25), however, their ionic conductivities are lower than 2m NaTFSI-PAE. The above electrolytes were further applied to the

symmetric PANI batteries. Those batteries didn't exhibit better cycling stability than 2m NaTFSI-PAE, possibly due to their stronger interactions with H_2O ⁵⁴ (Supplementary Fig. 26-28).”

new Supplementary Fig. 25. a, Electrochemical stabilities of 2m NaOTF-PAE, 2m NaClO₄-PAE, and 2m NaFSI-PAE, and 2m NaTFSI-H₂O. **b**, Ionic conductivities of 2m NaClO₄-PAE, 2m NaOTF-PAE, 2m NaFSI-PAE, and 2m NaTFSI-PAE.

new Supplementary Fig. 26. Electrochemical performances of all PANI batteries in 2m NaOTF-PAE. **a**, CV measurements. **b**, Cycling stability at a current density of 1 C (1 C=147 mA/g). **c**, Charge/discharge profiles in the first ten cycles at a current density of 1 C.

new Supplementary Fig. 27. Electrochemical performances of all PANI batteries in 2m NaClO₄-PAE. **a**, CV measurements. **b**, Cycling stability at a current density of 1 C (1 C=147 mA/g). **c**,

Charge/discharge profiles in the first ten cycles at a current density of 1 C.

new Supplementary Fig. 28. Electrochemical performances of all PANI batteries in 2m NaFSI-PAE. **a**, CV measurements. **b**, Cycling stability at a current density of 1 C (1 C=147 mA/g). **c**, Charge/discharge profiles in the first ten cycles at a current density of 1 C.

Comment #9:

Please include the actual FTIR spectra measured during cycling (corresponding to Fig. 3c,e) in the supplementary materials.

Response: As you suggested, we added actual FTIR spectra during cycling in **new Supplementary Fig. 17 and Fig. 19**.

(Supplementary Figure 17 and 19 in the revised Manuscript)

new Supplementary Fig. 17. Actual FT-IR spectra of PANI cathode at different voltage in 1st, 2nd cycles.

new Supplementary Fig. 19. Actual FT-IR spectra of PANI anode at different voltage in 1st, and 2nd cycles.

Comment #10:

Please indicate which mass are the specific capacity and energy density of all-PANI battery based on. Are they based on the total mass of active material in the cathode+anode? Or just based on one electrode? Similarly, please indicate in the method part that how were the specific capacity values calculated. Are they based on active material mass or the mass of the total electrode (AM+carbon+binder)?

Response: All calculations of specific capacity and energy density in this work are based on the mass of active materials in the cathodes. We added the calculation details of specific capacity and energy density in supporting information, which are also indicated in the figure 1 caption of the main text.

Considering the reviewer's comments, we have provided the calculated methods of specific capacity and energy in the revised Manuscript as below:

(Lines 692-697, Paragraph 1 of Page 19 in the revised Manuscript)

“Figure 1. Schematic figure of all-polymer aqueous sodium-ion batteries enabled by

polymer-aqueous electrolyte. d, Comparison with other all-organic ASIBs assembled with carbonyl derivatives anodes and PANI cathodes. **e,** Specific capacity (SC), energy density (ED in blue), capacity retention (CR), cycle number (CN), metal-free (MF) and electrolyte dilution (ED in red) of this work in comparison with advanced ASIBs. Both specific capacity, and energy density are calculated based on the mass of active materials in cathodes. MF, all organic electrodes: 1; single organic electrode: 0.5; all inorganic electrodes: 0.”

Comment #11:

Supplementary Fig. 14 exhibits a strong anodic peak at ~1.8 V in the first cycle but not in the subsequent cycles. Could the authors explain?

Response: We suppose the strong anodic peak at ~1.8 V originated from the decomposition of electrolyte during the SEI formation. To confirm the peaks, we conducted CV measurements with different organic electrodes in the 2m NaTFSI-PAE. In all cases, there is a strong anodic peak around 1.8 V, which proves this strong redox reaction didn't depend on the species of electrodes. The FTIR results of the PANI anode at different potentials also show that there are apparent TFSI⁻ signal around 1210 cm⁻¹ and 1150 cm⁻¹ in the first cycle. After 1.8 V, the TFSI⁻ signal becomes gradually weaker, and finally unapparent in the second cycle. The above results prove that the electrolyte is decomposed on the surface of the PANI anode, leading to the strong anodic peak at ~1.8 V. After forming stable SEI layers, the decomposition of electrolyte was suppressed, making the anodic peak at 1.8 V unapparent in the following cycles.

Considering the reviewer's comments, we have provided **new Supplementary Fig. 19-21 and Supplementary Fig. 22** and revised the following discussions:

(Lines 257-263, Paragraph 3 of Page 6 in the revised Manuscript)

“The anodic peak at ~1.8 V was observed in the first cycles in 2m NaTFSI-PAE, which could be attributed to forming a solid electrolyte interphase. CV measurements of other all-organic batteries in 2m NaTFSI-PAE confirmed that the anodic peak does not depend on the electrode species (**Supplementary Fig. 20, and 21**). After the first cycle, the symmetric PANI batteries exhibited a reversible redox reaction of PANI around 1.2 V without any other side reactions between 0.1-2.2 V (**Supplementary Fig. 22**).”

new Supplementary Fig. 19. Actual FT-IR spectra of PANI anode at different voltage in 1st, 2nd cycles.

new Supplementary Fig. 20. CV measurements with different organic electrodes in the 2m NaTFSI-PAE. a, PNFE||PANI. b, PTCDI||PANI.

new Supplementary Fig. 21. CV measurements with different organic electrodes in the 2m NaTFSI-PAE. a, NTCDA||PPy. b, PTCDI||PPy.

Supplementary Fig. 22. CV curves of all-PANI battery in 2m NaTFSI-PAE.

Comment #12:

The authors used XPS to characterize SEI on PANI anode after cycling. But the reviewer is curious if the authors have conducted depth profiling of the cycled PANI anode to confirm there is actually SEI on the surface? Basically, the XPS spectra should become similar to pristine PANI after ion sputtering which removes the surface (e.g., Na signal should be gone,).

Response: Many thanks for your kind reminder! According to your suggestions, we added XPS measurements of the anode after ion sputtering at different depths with pristine PANI as a reference. The results show that the intensity of Na1s increased within 1 min of ion sputtering and then decreased with the elongation of ion sputtering time. The increased Na1s signal is due to the

decomposition of residual NaTFSI on the anode surface. The decreased Na1s signal and increased C1s signal after 1 min of ion sputtering prove that the SEI is indeed on the PANI anode surface.

Considering the reviewer's comments, we have provided **new Figure 4 and new Supplementary Fig. 31** and revised the following discussions:

(Lines 302-315, Paragraph 2 of Page 7 in the revised Manuscript)

“For further understanding of the SEI structure, XPS of the anode surface with different ion sputtering times was conducted. With the sputtering time increasing, the intensity of O1s kept decreasing, but the intensity of Na1s and F1s increased within 1 min of ion sputtering and then decreased, indicating the SEI structures consisted of multiple components (**Fig. 4e**). Detailed analyses were conducted by fitting XPS spectra of N1s, O1s, F1s, and Na1s with 0 min, 1 min, 7 min ion sputtering. After 7 min sputtering, the –NH– groups of PANI appeared, proving the SEI layers grew on the PANI surfaces. Besides, the fitting results demonstrate the decrease of R-OCO₂Na, and the increase of NaOH, and NaF after ion sputtering (**Fig. 4d**). Although NaF may be generated from the decomposition of NaTFSI after 1 min of ion sputtering, the decomposed layers of NaTFSI should be removed after 1 min ion sputtering, which can be judged from the intensity change of byproduct C-SO_x in S2p (**Supplementary Fig. 31**). Then by comparing the ratio of different Na salt species in SEI layers by XPS of O1s, Na1s, and F1s (**Fig.4f**), we can determine the SEI structures shown in **Fig. 4g**, where R-OCO₂Na is the main component in SEI, accompanied with NaTFSI in the outer layer, NaOH, and NaF in the inner layer.”

new Figure 4. Electrochemical performance and solid electrolyte interphase. a, Galvanostatic charge-discharge curves of all-PANI battery at 20th, 50th, 500th, 1000th, 4000th cycle. **b**, Cycling performance and coulombic efficiency in 4800 cycles (381 days). **c**, SEM images of the polyaniline

surface after different cycles (pristine, after 10 and 100 cycles). **d**, XPS spectra (O1s, Na1s, F1s, and N1s) of SEI on polyaniline anode after different sputtering time (0, 1, 7mines). **e**, Intensities of different elements (O1s, Na1s, F1s, N1s, S1s, C1s) of SEI on after different sputtering time. **f**, Ratio of different Na salts of SEI on after different sputtering time (0, 1, 7mines). **g**, SEI structure on PANI anodes.

new Supplementary Fig. 31. a, Decomposition reaction of NaTFSI on PANI anode surface. **b**, Fitting of XPS spectra of C1s and S2P after ion sputtering with 0, 1, and 7 min.

Comment #13:

Please provide more detailed discussion of the structure/component of SEI, including a proposed structural scheme of SEI and formation mechanism. For example, does the SEI form due to the decomposition of PANI or polymer electrolyte or salt? Does different Na salt affect the formation of SEI? Since enabling the PANI anode is one of the major novelties of this work, a more detailed analysis is needed.

Response: To clarify the SEI component, we conducted in-depth XPS of our SEI and analyzed the formation of SEI in control samples including replacing PANI anode with active carbon and

replacing anion species. All above experiments revealed that PEGDME plays the most significant role in SEI formation, and the main component of SEI is R-OCO₂Na.

Considering the reviewer's comments, we have provided **new Supplementary Fig. 33-36** in the revised Manuscript and revised the following discussions:

(Lines 302-333, Paragraph 2 of Page 7 to Paragraph 1-3 of Page 8 in the revised Manuscript)

For further understanding of the SEI structure, XPS of the anode surface with different ion sputtering times was conducted. With the sputtering time increasing, the intensity of O1s kept decreasing, but the intensity of Na1s and F1s increased within 1 min of ion sputtering and then decreased, indicating the SEI structures consisted of multiple components (**Fig. 4e**). Detailed analyses were conducted by fitting XPS spectra of N1s, O1s, F1s, and Na1s with 0 min, 1 min, 7 min ion sputtering. After 7 min sputtering, the –NH– groups of PANI appeared, proving the SEI layers grew on the PANI surfaces. Besides, the fitting results demonstrate the decrease of R-OCO₂Na, and the increase of NaOH, and NaF after ion sputtering (**Fig. 4d**). Although NaF may be generated from the decomposition of NaTFSI after 1 min of ion sputtering, the decomposed layers of NaTFSI should be removed after 1 min ion sputtering, which can be judged from the intensity change of byproduct C-SO_x in S2p (**Supplementary Fig. 31**). Then by comparing the ratio of different Na salt species in SEI layers by XPS of O1s, Na1s, and F1s (**Fig. 4f**), we can determine the SEI structures shown in **Fig. 4g**, where R-OCO₂Na is the main component in SEI, accompanied with NaTFSI in the outer layer, NaOH, and NaF in the inner layer.

The SEI formation mechanism was further studied. First, the surface of active carbon (AC) in the half-cell was analyzed by XPS (**Supplementary Fig. 33**). The XPS results of AC didn't show a big difference from PANI, which means the PANI may not participate in the SEI formation. It is noteworthy that the intensity of Na1s in the 1st cycle is weak, but the structural transformation of the PANI anode from benzenoid (B) ring to quinonoid (Q) is apparent in FTIR spectra, indicating that the PANI anode may be reduced by protons from the H₂O (**Supplementary Fig. 19**). The XPS results of the PANI anode after ion sputtering also revealed that NaOH was generated in the SEI layer. Hence, although the PANI anode was not directly involved in SEI components, it may induce the generation of NaOH and facilitate the formation of SEI layers. Then, the effect of anion species on the SEI structure was further analyzed with the PANI anode after charge-discharge in 2m NaClO₄-PAE, and 2m NaOTf-PAE for 10, 50, and 200 cycles (**Supplementary Fig. 34-36**). Apart from 2m NaClO₄-PAE, the XPS results of PANI anodes are almost the same as those in 2m NaTFSI-PAE, indicating a similar surface composition of SEI attributed to PEGDME decomposition.

The above systematic studies revealed that PEGDME plays the most significant role in SEI formation, and the main component of SEI is R-OCO₂Na. Hence, our PAE not only improved the high electrochemical stability of H₂O, but also benefited SEI formation, which contributed to the cycling stability of all-PANI ASIBs.”

new Figure 4. Electrochemical performance and solid electrolyte interphase. d, XPS spectra (O1s, Na1s, F1s, and N1s) of SEI on polyaniline anode after different sputtering time (0, 1, and 7 mines). **e**, Intensities of different elements (O1s, Na1s, F1s, N1s, S1s, and C1s) of SEI on after different sputtering time. **f**, Ratio of different Na salts of SEI after different sputtering times (0, 1, and 7 mines). **g**, SEI structure on PANI anodes.

AC electrodes after different cycles

new Supplementary Fig. 33. XPS spectra (C1s, O1s, and Na1s) of SEI on AC electrodes after different cycles (pristine, 10th, 50th).

XPS of PANI electrodes after different cycle (2m NaOTf-PAE)

new Supplementary Fig. 34. XPS spectra (C1s, O1s, and Na1s) of SEI on polyaniline anode after different cycles (10th, 50th, and 200th) in 2m NaOTf-PAE electrolyte.

XPS of PANI electrodes after different cycle (2m NaClO₄-PAE)

new Supplementary Fig. 35. XPS spectra (C1s, O1s, and Na1s) of SEI on polyaniline anode after different cycles (10th, 50th, and 200th) in 2m NaClO₄-PAE electrolyte.

new Supplementary Fig. 36. XPS spectra (C1s, O1s, and Na1s) of SEI on polyaniline anode after different cycles (10th, 50th, and 200th) in 2m NaFSI-PAE electrolyte.

Comment #14:

EIS studies of the PANI anode were only conducted at the cycled state (charged to 2.2 V), but not at a discharged state. Please provide the EIS data when the cell is discharged to 0.1 V (or close to 0.1 V) after different cycles and corresponding interpretation.

Response: Many thanks for your important suggestions! We have added the EIS data when the cell is discharged to 0.1 V after 1, 50 and 100 cycles, which also didn't show a significant change, indicating that the SEI layer is stable during cycling.

Considering the reviewer's comments, we have provided EIS spectra (**new Supplementary Fig. 29**) when the cell is discharged to 0.1 V after different cycles in the revised Manuscript and revised the following discussions:

(Lines 289-291, Paragraph 2 of Page 7 in the revised Manuscript)

“Evident semicircles corresponding to R_{ct} and R_{SEI} appeared after the first cycle, and the charge transfer resistance significantly decreased, indicating the formation of continuous and dense solid-electrolyte interphase (SEI) layers⁵⁶.”

new Supplementary Fig. 29. a, Nyquist plots of all-PANI battery charged to 2.2 V after different cycles. **b,** Nyquist plots of the all-PANI battery charged to 0.1 V after different cycles.

Response to Reviewer #2&3:

Comment #0:

This paper reports a flexible aqueous sodium-ion battery (ASIB) featuring symmetric polyaniline (PANI) electrodes, and polyethylene glycol dimethyl ether (PEGDME) electrolyte to modulate the solvation layers and form a solid-electrolyte interphase. The assembled symmetric PANI ASIBs demonstrated a high capacity (139 mAh/g) and a stable and long cycle life (>4800 cycles with 92% capacity retention). Overall, the characterization of all-polymer aqueous batteries is systematically investigated, and the results look good. I think it should be published after minor revisions, as listed below.

Response: Thank you very much for the important comments and inspiring suggestions. We have conducted a suite of experiments to address all the reviewer's concerns. For convenience, the main revisions are discussed in the following point-to-point answers to the reviewer's questions and all main revisions are marked with **blue font** in the revised Manuscript.

Comment #1:

The manuscript claims that the H-O bond in the water is loose by adding PEGDME, reducing the activity of H₂O in the electrolyte. It seems the ether groups may play an essential role in improving the electrochemical stability of electrolytes. The manuscript only tested a single material in this study. It is unclear whether the same conclusions will remain plausible with other ether-rich polymers.

Response: According to your suggestions, we also measured the electrochemical stability and ionic conductivities of polymer-aqueous-electrolytes with polypropylene glycol (PPG). The results show that the electrochemical stability window of 2m NaTFSI-PPG-H₂O is almost the same as 2m NaTFSI-PEGDME-H₂O. However, its ionic conductivity is much lower due to PPG's high viscosity compared with PEGDME. The above experiments prove that other ether-rich polymers can also improve the electrochemical stability of electrolytes.

Considering the reviewer's comments, we have provided **new Supplementary Fig. 10** and revised the following discussions:

(Lines 128-130, Paragraph 3 of Page 3 in the revised Manuscript)

“Apart from PEGDME, we found other ethers containing polymer like polypropylene glycol (PPG) also can enlarge the electrochemical stability of aqueous electrolyte but shows lower ionic conductivities (Supplementary Fig. 10).”

new Supplementary Fig. 10. a, Electrochemical stabilities of 2m NaTFSI-^PPAE, and 2m NaTFSI-PAE. **b**, Ionic conductivities of 2m NaTFSI-PEGDME-H₂O, and 2m NaTFSI-PPG-H₂O.

Comment #2:

The manuscript tested low concentrations of salt only in this study. It would be better to see the concentration dependence of electrochemical stability as a high concentration is predicted to enhance the ionic conductivity.

Response: To figure out the concentration dependency of ionic conductivity. The maximum solubility of NaTFSI in PAE is 5m (**Fig. R3**), so we measured ionic conductivities with different concentrations (1m, 2m, 3m, 4m, and 5m) of NaTFSI in PAE. Unlike aqueous electrolyte, high salt concentration will cause lower ionic conductivities due to the sharp increase in viscosity. In addition, we also measured LSV of 4m NaTFSI-PAE, which didn't show an obvious change with double the salt concentration. In summary, 2m NaTFSI is the best concentration for PAE, judging from the electrochemical stability window and ionic conductivity.

Fig. R3. NaTFSI-PAE solution with different concentrations.

Considering the reviewer's comments, we have provided **new Supplementary Fig. 9** and revised the following discussions:

(Lines 124-128, Paragraph 3 of Page 3 in the revised Manuscript)

“Besides, the salt concentration dependencies of the ionic conductivities and the ESW of PAE were also studied (**Supplementary Fig. 9**). With the salt concentration increasing from 2m to 5m, the ionic conductivities decreased due to the sharply increasing viscosity. The ESW of 4m NaTFSI-PAE also didn't show an obvious increment compared 2m NaTFSI-PAE.”

new Supplementary Fig. 9. a, Ionic conductivities of 2m, 3m, 4m, and 5m NaTFSI-PAE. **b**, Electrochemical stabilities of 2m, and 4m NaTFSI-PAE.

Comment #3:

In Figure 4a, the capacity decreases significantly over the cycles, which is contrary to the results in Figure 4b. Please double check the data used.

Response: Thank you for your kind reminder. The previous charge-discharge profile in **new Figure 4a** contained 10 cycles of precycling data. We have already changed it to the proper results matching with **new Figure 4b**. And the precycling results are shown in **new Supplementary Figure 43**.

Considering the reviewer's comments, we have provided **new Figure 4 a, b and Supplementary Fig. 43** and revised the following discussions:

new Figure 4. Electrochemical performance and solid electrolyte interphase. a, Galvanostatic charge-discharge curves of all-PANI battery at 20th, 50th, 500th, 1000th, and 4000th cycle. **b,** Cycling performance and coulombic efficiency in 4800 cycles (381 days).

(Lines 472-474, Paragraph 1 of Page 12 in the revised Manuscript)

“The galvanostatic cycling test was conducted on a CT-4008T instrument (Shen Zhen NEWARE Electronic Co.), all batteries are precycled for 10 cycles before cycling measurements (Supplementary Fig. 43).”

new Supplementary Fig. 43. Electrochemical properties precycled for 10 cycles before cycling measurements. a, Galvanostatic charge-discharge curves of all-PANI battery at 1st, 3rd, 5th, 7th, and 10th. **b,** Cycling performance and coulombic efficiency in precycles.

Comment #4:

In line 182, it mentioned that the peak of S-N-S (879 cm^{-1}) periodically occurred in the figure.3c. But the range of wavenumber in fig.3c is from $1000\text{ to }1800\text{ cm}^{-1}$.

Response: Many thanks for your kind reminder. We have revised the range of wavenumber ($800\text{-}1800\text{ cm}^{-1}$) in **new Figure 3c and 3e**.

new Figure 3. Dual-ion doping mechanism of all-polymer aqueous sodium batteries with symmetric polyaniline electrodes. c, FTIR spectra of PANI cathode at different potentials in 2 charge-discharge cycles. e, FTIR spectra of PANI anode at different potentials in two charge-discharge cycles.

Comment #5:

In fig.3f, please explain why the difference between intensities of Na1s from two discharging cycles is relatively large.

Response: The difference between the intensities of Na1s from two discharging cycles is caused by the formation of solid-electrolyte interphase (SEI). In the first cycle, Na-doping of PANI is difficult to achieve, because sodium pernigranilate cannot exist when exposed to an aqueous environment. Although the Na1s intensity in the first cycle is weak, the apparent structure transformation from the benzenoid (B) ring to the quinonoid (Q) can be observed, which may be due to the reduction of the PANI anode by proton doping. This supposition was further proved by the XPS analysis of the PANI anode after ion sputtering, where the NaOH was observed in SEI layer. Apart from the proton doping of the PANI anode, the primary reactions happening on the anode are the decomposition of electrolyte and the formation of SEI. After SEI formation, the Na-doping of PANI becomes possible, leading to the intensities of Na1s becoming much higher in the second cycle.

Considering the reviewer's comments, we have provided **new Fig. 4** and **new Supplementary Fig. 19 and 33** in the revised Manuscript and revised the following discussions:

(Lines 316-324, Paragraph 2 of Page 8 in the revised Manuscript)

“The SEI formation mechanism was further studied. First, the surface of active carbon (AC) in the half-cell was analyzed by XPS (**Supplementary Fig. 33**). The XPS results of AC didn't show a big difference from PANI, which means the PANI may not participate in the SEI formation. It is noteworthy that the intensity of Na1s in the 1st cycle is weak, but the structural transformation of the PANI anode from benzenoid (B) ring to quinonoid (Q) is apparent in FTIR spectra, indicating that the PANI anode may be reduced by protons from the H₂O (**Supplementary Fig. 19**). The XPS results of the PANI anode after ion sputtering also revealed that NaOH was generated in the SEI layer. Hence, although the PANI anode was not directly involved in SEI components, it may induce the generation of NaOH and facilitate the formation of SEI layers.”

new Figure 4. Electrochemical performance and solid electrolyte interphase. e, Intensities of different elements (O1s, Na1s, F1s, N1s, S1f, and C1s) of SEI on after different sputtering time. **f**, Ratio of different Na salts of SEI after different sputtering times (0, 1, and 7mines). **g**, SEI structure on PANI anodes.

new Supplementary Fig. 19. Actual FT-IR spectra of PANI anode at different voltage in 1st, and 2nd cycles.

new Supplementary Fig. 33. XPS spectra (C1s, O1s, and Na1s) of SEI on AC electrodes after different cycles (pristine, 10th, and 50th).

Comment #6:

In fig.4a, although the all-PANI battery exhibited a high specific capacity at 1C, the voltage increased/decreased fast during charging/discharging process. Normally, a stable charge/discharge platform can ensure a constant and safe power supply for devices in practical applications. Could the author explain why there is no apparent platform during process?

Response: No apparent charge/discharge platform is a common issue for conductive polymer active materials. During the doping/dedoping process, the band gap of the conductive polymer usually changes quickly, which makes the voltage increase/decrease fast during the charging/discharging process. In the future, we will try to design novel polymeric electrode materials to address this issue. Besides, in practical applications, the batteries are usually charged/discharged under constant voltage, not constant current, which may also help avoid this issue.

Considering the reviewer's comments, we have provided the following discussions:

(Lines 351-354, Paragraph 1 of Page 9 in the revised Manuscript)

“Although conductive polymer active materials typically lack apparent charge/discharge platform due to the changing band gap during the doping/dedoping, the charge/discharge of batteries can be controlled under the constant voltage in the practical application.”

Comment #7:

In Page 6, it's mentioned that the SEI layers in ASIB originated from the reduction of electrolyte.

As the author claimed that PANI was firstly used as anode, is it also contributed to the formation of robust SEI?

Response: Thank you very much for your suggestions. We have a similar consideration to yours at the beginning. However, we cannot find proof that PANI is involved in SEI formation by FTIR, NMR, XPS, which didn't show a new chemical structure related to PANI. To further understand the role of PANI in SEI formation, we used active carbon as an anode to replace PANI and assemble the AC||PANI battery with PAE. After 10 and 50 cycles, we measured XPS of AC surface, which didn't show any obvious difference with PANI. Although PANI anode was not directly involved in SEI components, it induced the generation of NaOH in the SEI layers.

Considering the reviewer's comments, we have provided **new Fig. 4e, f, g**, and **Supplementary Fig. 19 and 33** in the revised Manuscript and revised the following discussions:

(Lines 316-324, Paragraph 2 of Page 8 in the revised Manuscript)

“The SEI formation mechanism was further studied. First, the surface of active carbon (AC) in the half-cell was analyzed by XPS (**Supplementary Fig. 33**). The XPS results of AC didn't show a big difference from PANI, which means the PANI may not participate in the SEI formation. It is noteworthy that the intensity of Na1s in the 1st cycle is weak, but the structural transformation of the PANI anode from benzenoid (B) ring to quinonoid (Q) is apparent in FTIR spectra, indicating that the PANI anode may be reduced by protons from the H₂O (**Supplementary Fig. 19**). The XPS results of the PANI anode after ion sputtering also revealed that NaOH was generated in the SEI layer. Hence, although the PANI anode was not directly involved in SEI components, it may induce the generation of NaOH and facilitate the formation of SEI layers.”

new Figure 4. Electrochemical performance and solid electrolyte interphase. e, Intensities

of different elements (O1s, Na1s, F1s, N1s, S1f, and C1s) of SEI on after different sputtering time. **f**, Ratio of different Na salts of SEI after different sputtering times (0, 1, and 7mines). **g**, SEI structure on PANI anodes.

new Supplementary Fig. 19 Actual FT-IR spectra of PANI anode at different voltage levels in 1st and 2nd cycles.

new Supplementary Fig. 33. XPS spectra (C1s, O1s, and Na1s) of SEI on AC electrodes after different cycles (pristine, 10th and 50th).